# Crystal structures of an E1–E2–ubiquitin thioester mimetic reveal molecular mechanisms of transthioesterification

Lingmin Yuan[1,2], Zongyang Lv[1,2], Melanie J. Adams [1] & Shaun K. Olsen [1,2✉]

E1 enzymes function as gatekeepers of ubiquitin (Ub) signaling by catalyzing activation and transfer of Ub to tens of cognate E2 conjugating enzymes in a process called E1–E2 transthioesterification. The molecular mechanisms of transthioesterification and the overall architecture of the E1–E2–Ub complex during catalysis are unknown. Here, we determine the structure of a covalently trapped E1–E2–ubiquitin thioester mimetic. Two distinct architectures of the complex are observed, one in which the Ub thioester (Ub(t)) contacts E1 in an open conformation and another in which Ub(t) instead contacts E2 in a drastically different, closed conformation. Altogether our structural and biochemical data suggest that these two conformational states represent snapshots of the E1–E2–Ub complex pre- and post-thioester transfer, and are consistent with a model in which catalysis is enhanced by a Ub(t)-mediated affinity switch that drives the reaction forward by promoting productive complex formation or product release depending on the conformational state.

[1] Department of Biochemistry & Molecular Biology and Hollings Cancer Center, Medical University of South Carolina, Charleston, SC, USA. [2] Department of Biochemistry & Structural Biology University of Texas Health Science Center at San Antonio, San Antonio, TX, USA. ✉email: olsens@uthscsa.edu

Post-translational modification of proteins by ubiquitin (Ub) is a means of regulating fundamental cellular processes including cell cycle control, DNA repair, signal transduction, and immunity[1–3] and requires the sequential interactions and activities of three enzymes (E1, E2, and E3)[4–6]. Ub can be attached to target proteins in a number of different ways including as a single molecule or as a variety of polymeric chains linked together through the seven lysine residues or N-terminal methionine of Ub[7,8]. The wide variety in the types of Ub modifications that occur on substrate proteins accounts for diversity in the functional outcomes of ubiquitination which include alterations of protein stability, localization, intermolecular interactions, and activity, and thereby underlies the role of ubiquitination as a central player in the regulation of cellular function[3,9].

E1 enzymes function as gatekeepers of Ub signaling by specifically catalyzing activation and transfer of Ub to tens of cognate E2 conjugating enzymes[4–6]. During E1-catalyzed activation, the C-terminus of Ub is adenylated and subsequently becomes linked via thioester bond to a catalytic cysteine residue on E1[10,11]. This is followed by the recruitment of E2 conjugating enzymes and the transfer of Ub from the E1 catalytic cysteine to the E2 catalytic cysteine in a process called E1–E2 thioester transfer (or transthioesterification)[12–14]. Previous structural studies have shown that Ub E1 is a multidomain enzyme in which each domain plays a distinct functional role in its three catalytic activities of adenylation, thioester bond formation, and transthioesterification[15–22]. Active and inactive adenylation domains (AAD and IAD) are responsible for the recruitment of Ub and harbor the catalytic machinery for adenylation of the C-terminus of Ub. The E1 Cys domain is split into two globular half domains (first and second catalytic cysteine half domains, FCCH and SCCH, respectively) with the FCCH domain playing a role in Ub recognition and the SCCH harboring the catalytic cysteine residue involved in Ub thioester bond formation. Lastly, the Ub-fold domain (UFD), which has recently been identified as a potential druggable site for cancer treatment[23] is involved in molecular recognition of E2s and subsequently in the transfer of Ub from E1 to E2.

Recent studies have revealed that E1s undergo large conformational changes that are required for its ability to catalyze adenylation, thioester bond formation, and transthioesterification[16,18,19,21,22,24–26]. Adenylation and thioester bond are catalyzed at a single location on the enzyme that is reconfigured for catalysis of these distinct chemical reactions via a network of complementary conformational changes[21,24]. Following adenylation of Ub, which occurs with the SCCH domain in an open conformation with the catalytic cysteine residue ~35 Å away from the active site, a ~125° rotation (or closure) of the SCCH domain brings the E1 catalytic cysteine into proximity of the Ub C-terminus where thioester bond formation occurs[21,24]. This is followed by reopening of the SCCH domain and subsequent recruitment and adenylation of a second Ub molecule to form a so-called doubly loaded complex since it has a stoichiometry of 1:2 with the Ub undergoing transfer to E2 (Ub(t)) linked via thioester bond to the E1 catalytic cysteine and a second Ub bound noncovalently to its adenylation domain as a Ub-adenylate intermediate (Ub(a))[14,27]. During transthioesterification, E2 is recruited to E1 with the UFD in a distal conformation resulting in a distance of ~25 Å between the E1 and E2 active sites[15,16]. The transition of the UFD from the distal to the proximal conformation bridges this gap and brings the E1 and E2 active sites in proximity to each other as occurs during transthioesterification[16,19,22]. While singly loaded E1 is capable of catalyzing transthioesterification, doubly loaded E1 is significantly more active though the structural basis for this is unknown[14,27]. Also unknown is the architecture of the doubly loaded E1 complex during transthioesterification, in particular

the Ub(t) molecule as all Ub E1–E2 structures determined to date lack Ub(t). These questions remain unanswered as several complexes and intermediates formed during E1–E2–Ub thioester transfer are transient and low affinity which has presented challenges to structural analyses.

To better understand the molecular basis of E1–E2–Ub transthioesterification, we sought to determine the first structure of a Ub E1–E2 complex that includes the Ub molecule undergoing transfer. Using an approach that involves the generation of a stable E2–Ub thioester intermediate mimetic and covalent trapping of the E1 and E2–Ub(t) mimetic active sites in proximity to each other, we here determine the structure of a S. cerevisiae E1 (Uba1)–E2(Cdc34)–Ub(t) complex. Remarkably, two distinct architectures of the complex are observed, one in which Ub(t) contacts the FCCH domain of Uba1 in an open conformation and another in which Ub(t) adopts a dramatically different closed conformation and contacts Cdc34. Our data suggest that these two conformational states represent snapshots of the E1–E2–Ub complex pre- and post-thioester transfer, and are consistent with a model in which catalysis is enhanced by a Ub(t)-mediated affinity switch that drives the reaction forward by promoting productive complex formation or product discharge depending on the conformational state. Collectively, our structural and biochemical studies reveal key insights into mechanisms of E1–E2–Ub transthioesterification that have long remained elusive.

## Results

**Trapping a doubly loaded E1–E2–Ub(a)–Ub(t) mimetic**. We sought to determine the structure of a doubly loaded E1–E2–Ub (a)–Ub(t) complex in order to gain a better understanding of the molecular basis of E1–E2–Ub transthioesterification, in particular, the role of Ub(t). We focused on the S. cerevisiae E1, Uba1, and the E2, Cdc34, because of the defined biological importance of this enzyme pair as a key regulator of the cell cycle and due to the fact that they are amenable to crystallization. Structural studies of doubly loaded E1–E2–Ub(a)–Ub(t) complexes have been hampered by the labile nature of thioester bonds, the low affinity of E1–E2 complexes, and the transience of the intermediate formed during transthioesterification. With regard to the lability of thioester bonds, a large number of structural studies have employed a strategy to circumvent this challenge in which the E2 catalytic cysteine is mutated to a lysine which is subsequently conjugated to Ub to form a stable mimetic that resembles the E2–Ub thioester intermediate[28–33]. We used a variation of this approach in which a residue in close proximity to the E2 catalytic cysteine is mutated to a lysine (Cdc34$^{A141K}$) for subsequent conjugation to Ub (Fig. 1a). The ability of the Cdc34$^{A141K}$–Ub complex to serve as a thioester mimetic (hereafter referred to as Cdc34–Ub(t)) is supported by a recent structural study of a trapped SUMO E2–SUMO-RING E3–substrate complex in which the E2–SUMO thioester complex was stabilized using a similar strategy[34]. Importantly, that the Cdc34–Ub(t) thioester mimetic retains its catalytic cysteine is key to overcoming the second challenge, namely the low affinity of Uba1–E2 interactions and transience of the intermediate formed during transthioesterification. Here, we employed a strategy that involves cross-linking of the Uba1 and Cdc34 catalytic cysteine residues (here in the context of Cdc34$^{A141K}$-Ub) that have been used to determine several previously reported E1–E2 structures in the absence of Ub (t) (Fig. 1a, Supplementary Fig. 1a)[16,18,22]. Our ability to generate Cdc34$^{A141K}$–Ub(t) in which Ub(t) is specifically conjugated to A141K, and to subsequently crosslink this complex to Uba1 specifically through their catalytic cysteine residues (Cys600 and Cys95, respectively) is shown in Fig. 1b. Importantly,

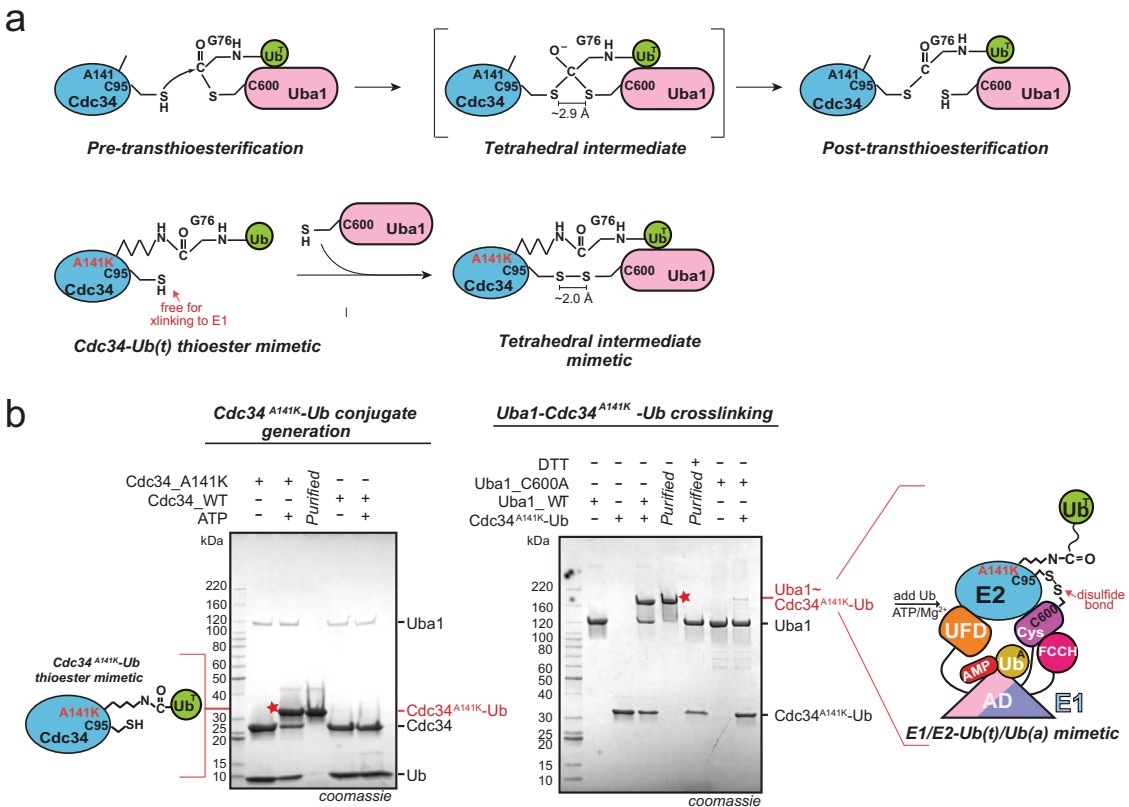

**Fig. 1 Strategy of generating E1/E2–Ub(t)/Ub(a) mimetic. a** Schematic of the E1–E2 thioester transfer intermediate and the mimetic used in this study. **b** Left, SDS-PAGE gel showing specific production of Cdc34[A141K]–Ub(t) conjugate in the presence of DTT. Right, Strategy to obtain Uba1/Cdc34[A141K]–Ub (t)/Ub(a). Gel images are representative of independent technical replicates (n = 2). Source data are provided as a Source Data file.

analytical gel filtration shows that purified Uba1–Cdc34–Ub(t) complex stably binds Ub in the presence of ATP•Mg[2+] but not in the absence of ATP•Mg[2+] or when an adenylation deficient mutant of Uba1 (D544A) is used (Supplementary Fig. 1b). This indicates that the covalently trapped complex retains key structural features required for enzyme function and that the resulting purified Uba1–Cdc34–Ub(t)/Ub(a) complex represents a mimetic of the intermediate formed during bona fide E1–E2–Ub transthioesterification that has been stabilized for structural studies.

**Two distinct architectures for the Uba1–Cdc34–Ub(t) complex.**
Reconstituted Uba1/Cdc34–Ub(t)/Ub(a)/ATP•Mg[2+] complex was subjected to crystallization trials and after extensive refinement, diffraction quality crystals yielding complete data sets were obtained. Molecular replacement yielded an initial solution with eight molecules of Uba1 and Cdc34 in the crystallographic asymmetric unit (Supplementary Fig. 1c). Inspection of the resulting electron density maps validated the placement of the eight copies of Uba1 and Cdc34 and revealed unambiguous electron density for eight copies of Ub(t) from the Cdc34–Ub(t) thioester mimetic into which Ub(t) was manually placed (Supplementary Fig. 2a, b). Despite the crystals having been grown in the presence of excess Ub and ATP•Mg[2+], electron density corresponding to Ub(a) was not observed in any of the Uba1–Cdc34–Ub(t) complexes. Further analysis reveals that this is due to crystal packing effects that perturb the orientation of the FCCH domain such that it partially occludes the Ub(a) binding site on the AAD of Uba1 and not due to structural differences or perturbations in the Ub(a)-binding site itself (Supplementary Fig. 2c). The ATP binding pocket of seven Uba1–Cdc34–Ub(t) complexes harbored strong electron density into which AMP

(but not the β and γ phosphates of ATP) could unambiguously be placed. The final Uba1–Cdc34–Ub(t) model was refined to R/R_free values of 0.203/0.246 with excellent geometry at a resolution of 3.43 Å (Supplementary Table 1).

Much to our surprise, two distinct architectures of the Uba1–Cdc34–Ub(t) thioester mimetic complex are observed in the crystallographic asymmetric unit, with the differences largely centered on the positioning of Ub(t) relative to a fixed Uba1–Cdc34 module (Fig. 2, Supplementary Fig. 1c). Previous structures have shown that E2–Ub(t) complexes adopt an array of conformations ranging from open in which Ub(t) extends away from the E2 active site, backbent in which Ub(t) folds backward and engages in contacts with the back surface of E2, and closed in which the Ub(t) engages in contacts to the front surface of the E2 (refs. [35–37]). In four copies of the Uba1–Cdc34–Ub(t) complex Ub(t) adopts an open conformation similar to E2–Ub(t)[OPEN] structures, extending away from the Uba1 and Cdc34 active sites where it engages in a network of contacts to the FCCH domain on the front of Uba1 (Fig. 2a). Interestingly, the positioning of Ub(t) in the Uba1–Cdc34–Ub(t)[OPEN] complex is nearly identical to that observed in the doubly loaded Uba1–Ub(t)/Ub(a) complex determined in the absence of E2 (PDB: 4NNJ).[17] with a total of ~800 Å² of Ub(t) surface area buried at the Ub(t)[OPEN]/FCCH domain interface (Fig. 2). With respect to the active sites, the catalytic cysteine of Uba1 (Cys600), catalytic cysteine of Cdc34 (Cys95) and Gly76 of Ub(t)[OPEN] are all in proximity to each other, as anticipated, with distances between the γ-sulfur atoms of the E1 and E2 catalytic cysteines and the carbonyl carbon of Gly76 of Ub measuring 4.1 Å and 4.6 Å, respectively (Fig. 2a, Supplementary Fig. 2d). Only slight structural rearrangements would place the relevant atoms in position for catalysis suggesting that our Uba1–Cdc34–Ub(t)[OPEN] structure approximates the

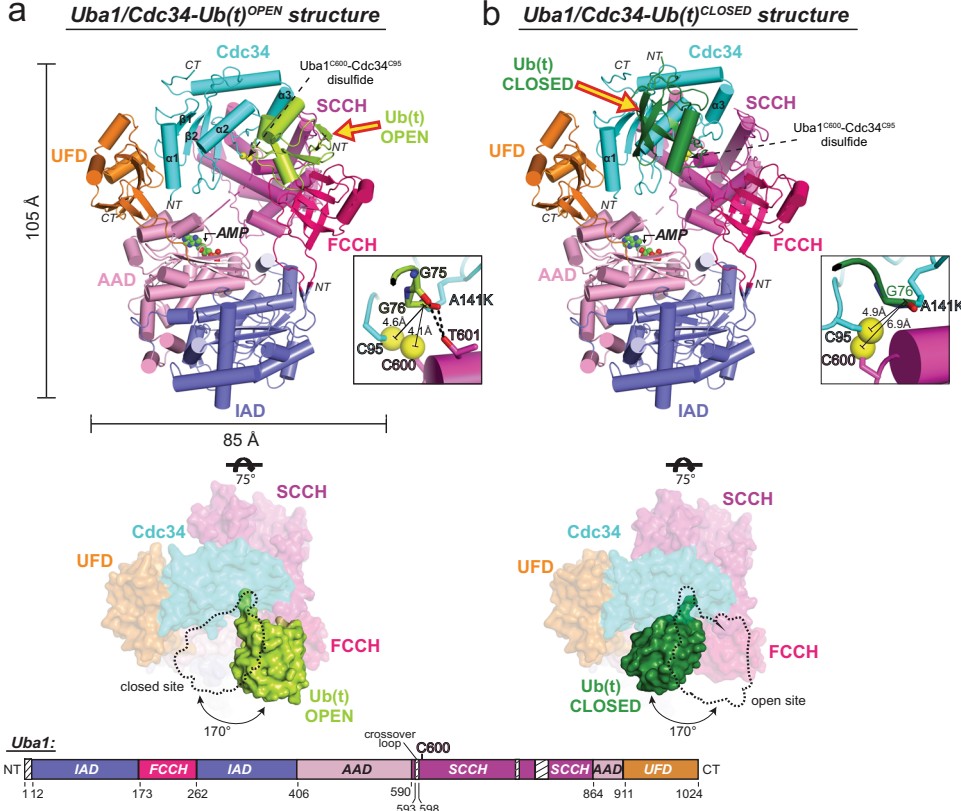

**Fig. 2 Overall architecture of Uba1/Cdc34^A141K–Ub(t)^OPEN and Uba1/Cdc34^A141K–Ub(t)^CLOSED structures. a** Top, Side view of the overall Uba1/Cdc34^A141K–Ub(t)^OPEN structure. Uba1 domains are colored and labeled, Cdc34^A141K is colored cyan, Ub(t)^OPEN is colored limon. Insets to the right of the structures highlight the relative distances among Uba1 Cys600, Cdc34 Cys95, and Ub(t) Gly76 within the active sites. Middle, The structure is shown from the top down and the fixed Uba1-Cdc34 module is shown as a semitransparent surface representation. Ub(t)^OPEN is shown as a nontransparent surface representation and the Ub(t)^CLOSED binding site is shown with a dashed line to allow for comparison. **b** Top, Side view of the overall Uba1/Cdc34^A141K–Ub(t)^CLOSED structure, Uba1 domains are colored and labeled, Cdc34^A141K is colored cyan, Ub(t)^CLOSED is colored forest. The right side illustrations show the relative distances among Uba1 Cys600, Cdc34 Cys95, and Ub(t) Gly76 catalytic sites. Middle, The structure is shown from the top down and the fixed Uba1-Cdc34 module is shown as a semitransparent surface representation. Ub(t)^CLOSED is shown as a nontransparent surface representation and the Ub(t)^OPEN binding site is shown with a dashed line to allow for comparison. Bottom, Schematic representation of Uba1 construct with domains colored as above. Regions of disorder are indicated with a hatched box. Amino acid number is indicated on the bottom.

complex as it exists just prior to bona fide E1–E2–Ub transthioesterification.

In stark contrast to the Uba1–Cdc34–Ub(t)^OPEN complex, Ub(t) adopts a drastically different closed conformation in the other four copies of the Uba1–Cdc34–Ub(t) complex in the asymmetric unit, projecting backward from the Uba1 and Cdc34 active sites and engaging in extensive contacts with the front surface of Cdc34 (Fig. 2b). The Ub(t)^CLOSED/Cdc34 interface observed in the Uba1–Cdc34–Ub(t)^CLOSED structure is similar to RING E3-activated closed E2–Ub(t) complexes[28–31,38,39] with a total of ~750 Å² of Ub(t) surface area buried at the interface. In both the Uba1–Cdc34–Ub(t)^OPEN and Uba1–Cdc34–Ub(t)^CLOSED complexes, electron density corresponding to the isopeptide bond between A141K of Cdc34 and Gly76 of Ub(t) and the disulfide bonds between the Uba1 and Cdc34 active site cysteines used to stabilize the Uba1–Cdc34 interface are all well-ordered (Supplementary Fig. 2b, e).

With respect to the active sites, the Uba1 and Cdc34 catalytic cysteines and Gly76 of Ub(t)^CLOSED are in close proximity, with distances between the γ-sulfur atoms of the E1/E2 catalytic cysteines and the carbonyl carbon of Ub Gly76 measuring 6.9 Å and 4.9 Å, respectively (Fig. 2b, Supplementary Fig. 2e). Compared to the Uba1–Cdc34–Ub(t)^OPEN complex, the carbonyl carbon of Ub Gly76 is farther away from the γ-sulfur atom of the

E1 catalytic cysteine and significantly closer to the γ-sulfur atom of the E2 catalytic cysteine. In fact, the distance between the γ-sulfur atom of the E2 catalytic cysteine and the carbonyl carbon of Ub Gly76 can be reduced to only 3.3 Å with a simple rotamer change of Cys95 side chain. The remaining distance could easily be reduced by minor conformational changes that would place the relevant atoms in positions compatible with a Cdc34–Ub thioester bond. Remarkably, and as will be described in detail below, analysis of our Uba1–Cdc34–Ub(t)^OPEN and Uba1–Cdc34–Ub(t)^CLOSED complexes suggests that we have captured catalytic snapshots of the complex just prior to and post Uba1–Cdc34–Ub transthioesterification.

With respect to the fixed Uba1–Cdc34 module, in all copies of the Uba1–Cdc34–Ub(t) thioester mimetic complex, Uba1 and Cdc34 engage each other in a similar manner with Cdc34 sandwiched in the canyon between the UFD and SCCH domains of Uba1. The majority of contacts to the UFD are mediated via N-terminal helix α1 and the β1–β2 loop of Cdc34 whereas the majority of contacts to the SCCH domain are mediated by the α2–α3 loop and α3 of Cdc34 (Fig. 2). These interfaces are similar to those observed in the previous E1–E2 structures determined in the absence of Ub(t), including Uba1–Cdc34[16,18,22] so details will not be described here. Uba1 undergoes large conformational changes that play an important

role in its ability to catalyze adenylation, thioester bond formation, and transthioesterification. Most notably, the SCCH domain adopts open and closed conformations related to each other by a ~125° rotation[21,24] and the UFD adopts proximal and distal conformations related to each other by ~25° (refs. [15,16,20]). In all copies of the Uba1–Cdc34–Ub(t) complex the SCCH domain of Uba1 adopts the open conformation that positions the Uba1 catalytic cysteine directly across from the UFD, and the UFD adopts a proximal conformation that places the Cdc34 active site adjacent to the Uba1 active site where electron density for the disulfide bond between Cys600 of Uba1 and Cys95 of Cdc34 that stabilizes the complex is evident (Fig. 2, Supplementary Fig. 2d, e).

**Uba1–Cdc34/Ub(t)$^{OPEN}$ interface: pre-transfer conformation**. Analysis of the Uba1/Cdc34–Ub(t)$^{OPEN}$ structure reveals three distinct interfaces between Ub(t)$^{OPEN}$ and the fixed Uba1–Cdc34 module that altogether serve to stabilize Ub(t) in the open conformation (Fig. 3, Supplementary Fig. 3a). The largest interface is between the β-grasp domain of Ub(t)$^{OPEN}$ and the FCCH domain of Uba1 where Leu8, Ile44, and His68, from the so-called Ile44 hydrophobic patch of Ub(t)$^{OPEN}$ engage in an extensive network of contacts with residues from the FCCH domain (Fig. 3a, b, d). Ile44 of Ub(t)$^{OPEN}$ is at the center of this network where it engages in van der Waals contacts to Thr196 and Phe236 of the FCCH domain. Surrounding this, His68 of Ub(t)$^{OPEN}$ participates in hydrogen bonds to Asp188, Glu190, and Thr196 of the FCCH domain and Leu8 of Ub(t)$^{OPEN}$ engages in van der Waals contacts to Leu198 of the FCCH domain. Further reinforcing this interface, the backbone carbonyl oxygens of Ala46 and Gly47 of Ub(t)$^{OPEN}$ engage in hydrogen bonds to the amide nitrogens of Gly234 and Phe236. In addition, Gln49 of Ub(t)$^{OPEN}$ engages in van der Waals contacts to Phe236 of the FCCH domain, and Lys6 of Ub(t)$^{OPEN}$ (K6R) engages in a pair of hydrogen bonds to Asp188 of the FCCH domain. The Ub(t)$^{OPEN}$/SCCH domain interface is comparatively smaller and primarily involves hydrogen-bond interactions. Specifically, Glu705 of the SCCH domain participates in hydrogen bonds with the Thr7 side-chain hydroxyl and Thr9 backbone amide of Ub(t)$^{OPEN}$, and Asn703 of the SCCH domain engages in hydrogen bonds to the backbone carbonyl of Glu34 of Ub(t)$^{OPEN}$ (Fig. 3a, b).

The third interface between Ub(t)$^{OPEN}$ and the fixed Uba1–Cdc34 module of the complex involves the C-terminus of Ub(t)$^{OPEN}$ and residues within and surrounding the Uba1 and Cdc34 active sites (Fig. 3a, b). Notably, residues from the α2–α3 loop of Cdc34 (amino acids 134–142) and H17 (residues 599–604) from Uba1 form the sides of a groove that guides residues 74–76 of the Ub(t) C-terminus to the E1/E2 active sites through a series of intermolecular interactions. Here, Leu73 of Ub(t)$^{OPEN}$ engages in van der Waals contacts to Leu599 of Uba1, and the side chain of Arg74 of Ub(t)$^{OPEN}$ participates in hydrogen bonds to the backbone carbonyl oxygens of Asp134 and Pro135. Gly75 of Ub(t)$^{OPEN}$ participates in backbone-mediated hydrogen bonds to Asn136 and Ser139 of Cdc34, and in van der Waals contacts to Leu131 of Cdc34 and Gly76 of Ub(t)$^{OPEN}$ participates in a hydrogen bond with Thr601 of Uba1 and is in proximity to the side chain of Ser139 of Cdc34. Importantly, the γ-sulfur atoms of the Uba1 and Cdc34 catalytic cysteines are within 4.1 Å and 4.6 Å of the carbonyl carbon of Gly76 of Ub(t), respectively (Fig. 2a, Supplementary Fig. 2d). Beyond the E1 and E2 active sites, the groove accommodating the Ub(t)$^{OPEN}$ C-terminus is obstructed by Tyr89 and Ala141 of Cdc34 and Arg603 and Ser604 of Uba1, further locking the Ub C-terminus in place and providing a platform for catalysis on which the Uba1 and Cdc34 catalytic cysteines as well as Gly76 of Ub(t)$^{OPEN}$ reside (Supplementary Fig. 3b, Left, Middle).

As discussed above, residues from the α2–α3 loop of Cdc34 play an important role in transthioesterification, and previous studies have shown that this loop is also crucial for isopeptide bond catalysis[28,40,41]. Notably, this loop has a tendency to be poorly ordered and an asparagine residue (Asn87 in Cdc34) from the very highly conserved HPN motif of E2s participates in intramolecular hydrogen bonds with residues from the α2–α3 loop to stabilize its conformation for catalysis of isopeptide bond formation[38,40]. In our Uba1/Cdc34–Ub(t)$^{OPEN}$ structure, the side chain of Asn87 of Cdc34 participates in three intramolecular hydrogen bonds to backbone atoms of Asn136 and Ser139 (Supplementary Fig. 3b, Right). Interestingly, our structure reveals that the α2–α3 loop is not only stabilized by intramolecular interactions during E1–E2 Ub transthioesterification but also by intermolecular interactions to Uba1. A highly conserved proline residue in the α2–α3 loop of Cdc34 (Pro140) participates in a network of van der Waals contacts to a hydrophobic patch on the SCCH domain of Uba1 formed by Phe605, Leu691, Phe695, and Phe707 and Asn138 of Cdc34 is within hydrogen-bonding distance of Lys712 of the Uba1 SCCH domain (Supplementary Fig. 3b). Both sets of contacts have previously been demonstrated to play an important role in E1–E2 Ub transthioesterification activity for other E1–E2 pairs[16,18,22].

Although the Ub(t)$^{OPEN}$ C-terminus was disordered in the Uba1–Ub(t)/Ub(a) doubly loaded structure that was previously determined in the absence of E2, contacts at the Ub(t)$^{OPEN}$/FCCH domain and Ub(t)$^{OPEN}$/SCCH domain interfaces are highly similar to those observed in our Uba1/Cdc34–Ub(t)$^{OPEN}$ structure (Fig. 3d)[17]. This, together with the observation that the γ-sulfur atoms of the Uba1 and Cdc34 catalytic cysteines are within distances of the carbonyl carbon of Gly76 of Ub where only slight structural rearrangements would place the relevant atoms in position for nucleophilic attack of the Gly76 carbonyl carbon by the γ-sulfur atom of the catalytic cysteine of Cdc34 as occurs during bona fide catalysis, suggests that we have captured a snapshot of the Uba1/Cdc34–Ub(t)$^{OPEN}$ complex that approximates the pre-transthioesterification state. The side chains of Ser139 of Cdc34 and Thr601 of Uba1 are in proximity to the carbonyl oxygen of Ub(t)$^{OPEN}$ Gly76 where they could form part of the oxyanion hole that stabilizes the negatively charged transition state during catalysis, in addition to their role in positioning the Ub(t) C-terminus (Supplementary Fig. 3b). Also worth noting is that the N-terminus of Uba1 H17 is in proximity to the carbonyl oxygen of Ub(t)$^{OPEN}$ where the positive electrostatic potential from the helix dipole could also contribute to transition state stabilization (Supplementary Fig. 3c). Side chains that would be capable of deprotonating the incoming Cys95 nucleophile during catalysis are not evident in the structure. Mutational analysis shows that substitution of individual residues from the FCCH and SCCH domains of Uba1 involved in contacts to Ub(t)$^{OPEN}$ have only a subtle effect on Uba1–Cdc34 Ub transthioesterification. In contrast, mutation of residues within the active site of Uba1 (T601V, T601V/N703D/E705K) as well as residues from the α2–α3 loop of Cdc34 (Asp134, Asn136, Ser139) results in moderate to severe reduction in Uba1–Cdc34 Ub transthioesterification activity consistent with a role in promoting the reaction (Fig. 3c, Supplementary Fig. 4).

As noted in previous studdies, immobilization of the thioester alone can provide marked increase in reactivity and in the case of ester hydrolysis reducing the conformational freedom of the ester and nucleophile results in a greater than 50,000 fold rate increase[40,42,43]. Analysis of our structures reveals that in the case of isoenergetic E1–E2 Ub transthioesterification, this conformational restriction is achieved by: (1) contacts involving the E1/E2 groove that guides the Ub C-terminus into the active site, (2) by

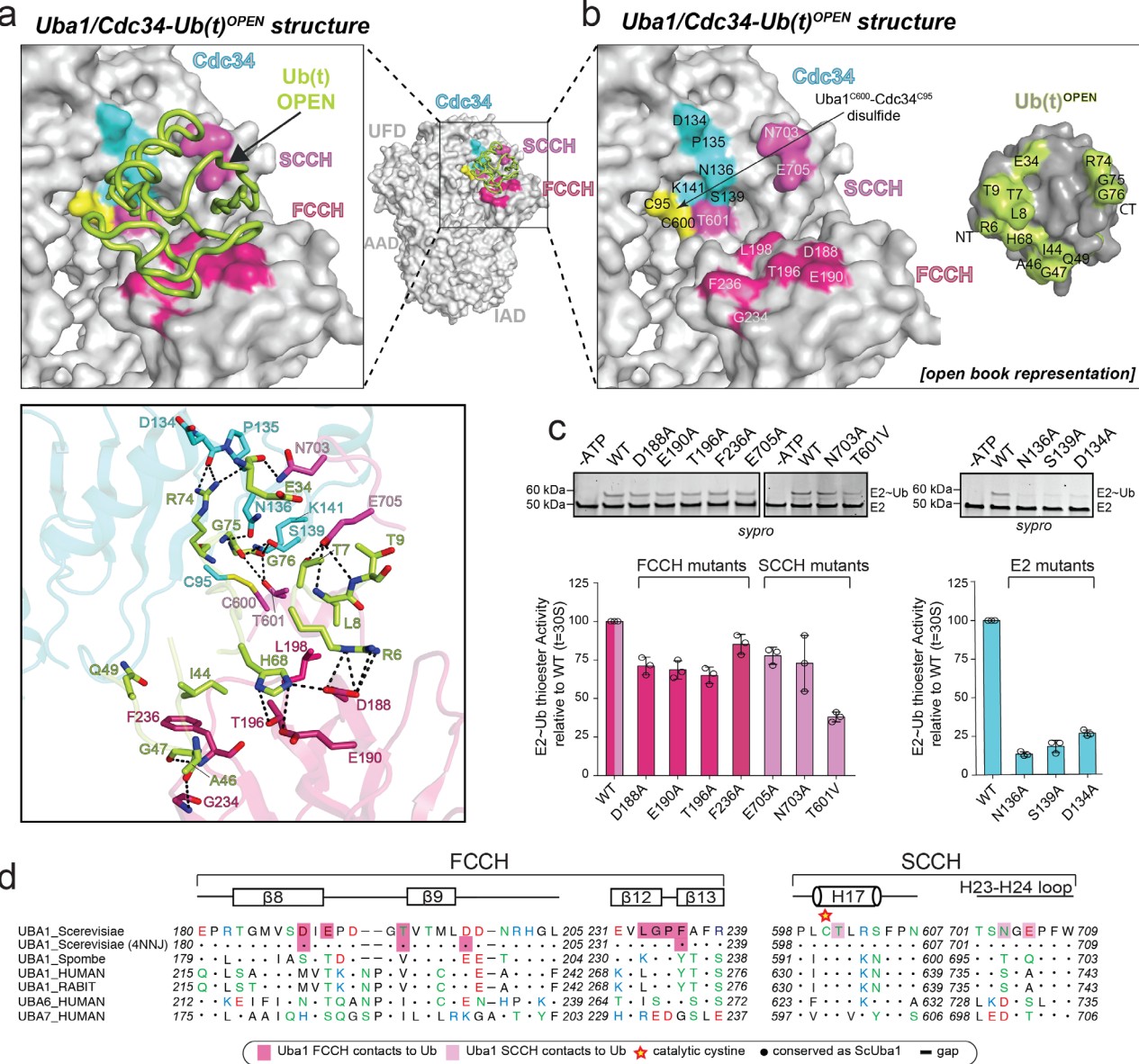

**Fig. 3 Interfaces of Uba1/Cdc34^A141K–Ub(t)^OPEN structure. a** Top, Uba1/Cdc34^A141K–Ub(t)^OPEN structure with surfaces of Uba1 domains and Cdc34 gray with only contacting residues colored. Ub(t) is represented as loop and colored limon. Bottom, Network of contacts between Uba1, Cdc34, and Ub(t) with involved residues shown as sticks with red oxygen atoms, blue nitrogen atoms, and yellow sulfur atoms. Hydrogen bonds are indicated by dashed lines. **b** Surface representations of Uba1, Cdc34, and Ub(t) shown in open book representation with residues in the interface colored hot pink (FCCH domain), light magenta (SCCH domain), cyan (E2), and limon (Ub(t)). **c** E1–E2 thioester transfer assays of the indicated mutants for Uba1 FCCH and SCCH domains. Data are represented by mean ± SD with three independent technical replicates labeled above and individual replicates shown as black circles. Gel images are representative of independent technical replicates (n = 3). Source data are provided as a Source Data file. **d** Sequence alignment of Uba1s FCCH and SCCH across different species with Uba1's secondary structure cartoon shown above sequence. Shaded boxes indicate Uba1 residues that interact with the Ub(t).

stabilization of the conformation of the α2–α3 loop of E2 through both intramolecular (primarily with Asn from the HPN motif) and intermolecular (with the SCCH domain of Uba1) interactions, and (3) termination of the E1/E2 groove and formation of the platform for catalysis on which the Uba1 and Cdc34 catalytic cysteines as well as Gly76 of Ub(t)^OPEN reside. As will be discussed in more detail below, we also posit that restriction of Ub(t) conformational variability through contacts to Uba1 that promote the Ub(t)^OPEN conformation promotes E1–E2 Ub transthioesterification by reducing steric clashes as E2 approaches the E1 active site during nucleophilic attack of the Uba1–Ub(t) thioester bond and by helping to maintain the SCCH domain in the open conformation.

**Uba1–Cdc34/Ub(t)^CLOSED interface: post-transfer conformation.** In contrast to the Uba1/Cdc34–Ub(t)^OPEN structure in which Ub(t) extends away from the Uba1/Cdc34 active sites and primarily interacts with the FCCH and SCCH domains of Uba1, analysis of the Uba1/Cdc34–Ub(t)^CLOSED structure reveals that Ub(t) instead primarily interacts with Cdc34 (Fig. 4a, b, d; Supplementary Fig. 5a). In the Uba1/Cdc34–Ub(t)^CLOSED structure, interactions between the β-grasp domain of Ub(t) and the FCCH and SCCH domain of Uba1 that stabilize Ub(t) in an open conformation in the Uba1/Cdc34–Ub(t)^OPEN structure are absent and instead the Ub(t)^CLOSED β-grasp domain is located proximal to the so-called crossover helix (α2) of Cdc34 where a network of interactions occurs that stabilize the Ub(t) closed conformation

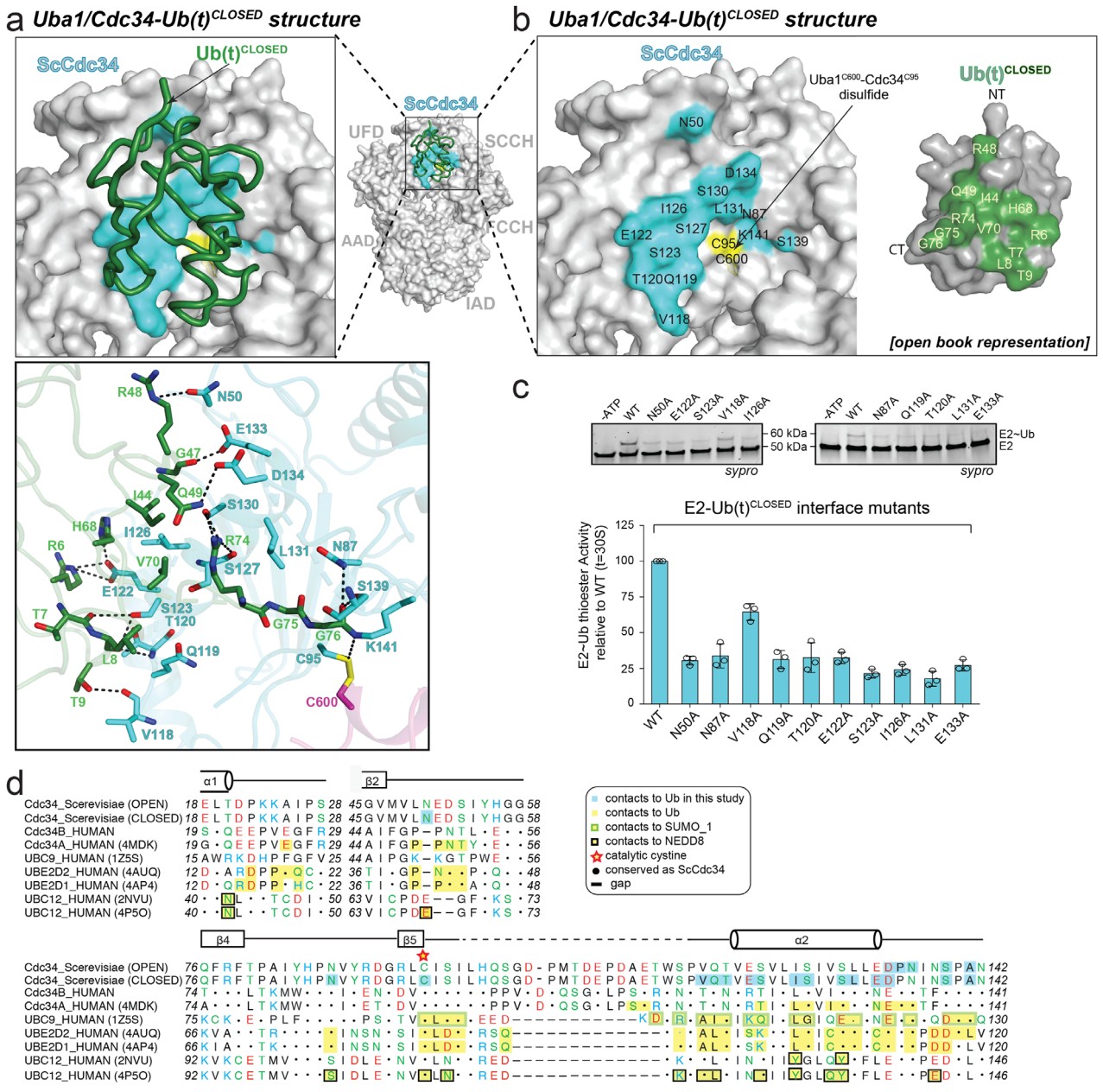

**Fig. 4 Interfaces of Uba1/Cdc34^{A141K}–Ub(t)^{CLOSED} structure. a** Top, Uba1/Cdc34^{A141K}–Ub(t)^{CLOSED} structure with surfaces of Uba1 domains and Cdc34 gray with only contacting residues colored. Ub(t) is represented as loop and colored green. Bottom, Network of contacts between Cdc34 and Ub(t) with involved residues shown as sticks with red oxygen atoms, blue nitrogen atoms, and yellow sulfur atoms. Hydrogen bonds are indicated by dashed lines. **b** Surface representations of Uba1, Cdc34, and Ub(t) shown in open book representation with residues in the interface colored cyan (E2) and green (Ub(t)). **c** E1–E2 thioester transfer assays of the indicated mutants for Cdc34. Data are represented by mean ± SD with three independent technical replicates labeled above and individual replicates shown as black circles. Gel images are representative of independent technical replicates (*n* = 3). Source data are provided as a Source Data file. **d** Sequence alignment of E2s across different species with Cdc34's secondary structure cartoon shown above sequence. Shaded boxes indicate E2s residues that interact with the Ub(t), Nedd8 or SUMO_1.

(Figs. 2, 4, Supplementary Fig. 5b). At the center of this interface, Leu8, Ile44, His68, and Val70 from the Ile44 hydrophobic patch of Ub(t)^{CLOSED} engages in a network of van der Waals interactions with Ile126, Ser127, and Ser130 of Cdc34. This central network of hydrophobic interactions is surrounded on either side by networks of predominantly hydrogen-bond mediated interactions. On one side, the backbone carbonyl oxygens of Thr7, Leu8, and Thr9 from Ub(t)^{CLOSED} engage in hydrogen bonds to Cdc34 Val118, Thr120, and Ser123 and His68 and Lys6 (K6R) engage in hydrogen bonds to Cdc34 Glu122. On the other side of the central hydrophobic network, Gln49 of Ub(t)^{CLOSED} engages in polar contacts to Ser130 of Cdc34 and Arg74 from the C-terminus of Ub(t)^{CLOSED} engages in hydrogen bonds to the backbone carbonyl oxygen of Ser127 and the side-chain hydroxyl of Ser130.

Mutations at this interface including N50A, N87A, Q119A, T120A, E122A, S123A, I126A, L131A, and E133A result in a significant decrease in Uba1–Cdc34-Ub transthioesterification activity suggesting an important role for this interface in catalysis that will be discussed in more detail below (Fig. 4a–c;

Supplementary Fig. 5c). Thermal shift assay shows that all of the Cdc34 mutants tested, with the exception of N87A, have similar melting temperatures as the wild type, which suggests these Cdc34 mutations do not affect the stability of the protein. The lower melting temperature observed for the Cdc34 N87A mutant relative to wild type (Supplementary Fig. 5d) suggests that its significantly diminished transthioesterification activity is at least partially due to structural defects resulting from the lack of the aforementioned intramolecular interactions with the α2–α3 loop of Cdc34. Lastly, it is worth noting that the interface between Ub(t) and Cdc34 observed in our Uba1/Cdc34–Ub(t)$^{CLOSED}$ complex is highly similar to previously determined closed E2–Ub structures in the presence and absence of E3 ligase[22,28–31,38,39] that are thought to be primed for discharge of Ub(t) onto target lysines during catalysis of isopeptide bond formation (Fig. 4d, Supplementary Fig. 5b, Supplementary Fig. 6).

In the Uba1/Cdc34–Ub(t)$^{CLOSED}$ structure, the extensive network of contacts between Arg74 and Gly75 of the Ub(t) C-terminus and residues Asp134, Pro135, Asn136, and Ser139 from the α2-α3 loop of Cdc34 that were observed in the Uba1/Cdc34–Ub(t)$^{OPEN}$ structure and validated to be important for transthioesterification are nearly all absent (Fig. 3a, c; Supplementary Fig. 3b; Fig. 4a, b). Instead, the Ub(t)$^{CLOSED}$ C-terminus adopts a significantly different conformation that accommodates placement of the β-grasp domain of Ub(t) at a totally different location on the fixed Uba1–Cdc34 module of the complex. In the Uba1/Cdc34–Ub(t)$^{CLOSED}$ structure, the carbonyl carbon of Ub Gly76 is located 4.9 Å away from the γ-sulfur atom of the Cdc34 active site cysteine (Cys95), and a simple rotamer change of Cys95 side chain could reduce this distance to 3.3 Å. The remaining distance could easily be reduced by minor conformational changes that would place the relevant atoms in positions compatible with a Cdc34–Ub thioester bond. In contrast, the γ-sulfur atom of the Uba1 active site cysteine (Cys600) is nearly 7 Å away from the carbonyl carbon of Ub Gly76 and more substantial structural rearrangements would need to occur in order for atoms to be positioned such that they are compatible with a Uba1–Ub thioester bond (Fig. 2b, Supplementary Fig. 2e). Altogether: (1) the closer proximity of the Cdc34 catalytic cysteine to the Ub C-terminus, (2) the lack of contacts between the Ub C-terminus and the catalytically important α2–α3 loop of Cdc34, and (3) the complete loss of an interface between Ub(t) and Uba1 and formation of a distinct network of contacts with Cdc34, lead us to surmise that our Uba1/Cdc34–Ub(t)$^{CLOSED}$ structure represents the complex as it exists post-transthioesterification and prior to product (Cdc34–Ub) release.

**An affinity switch mechanism for transthioesterification.** We next set out to place our Uba1/Cdc34–Ub(t)$^{OPEN}$ and Uba1/Cdc34–Ub(t)$^{CLOSED}$ structures in the context of previously determined E1 and E1–E2 structures to obtain a more complete picture of the mechanisms of Uba1-catalyzed reactions (Fig. 5). First, docking of Cdc34 (PDB: 6NYA)[22] onto the UFD of the doubly loaded Uba1-Ub(t)/Ub(a) structure (PDB: 4NNJ)[17] in which the UFD is observed in a distal conformation shows that the E1 and E2 active sites are indeed more than 18 Å away from each other (Fig. 5a, b, Left; Supplementary Fig. 7a). Comparison of the doubly loaded Uba1-Ub(t)/Ub(a)/Cdc34 model to our Uba1/Cdc34–Ub(t)$^{OPEN}$ structure reveals a 23° rotation of the UFD that as expected brings the E1 and E2 active sites in proximity along with a disordering of a flexible loop in the Uba1 SCCH domain termed the Cys cap that further exposes the Uba1–Ub thioester bond for nucleophilic attack by the Cdc34 catalytic cysteine. In contrast, the FCCH and SCCH domains exhibit only slight differences in conformation (Fig. 5a, b;

Supplementary Fig. 7a–c, Left, Middle). In our Uba1/Cdc34-Ub(t)$^{OPEN}$ structure, the β-grasp domain of Ub(t) sits on the FCCH and SCCH domains of Uba1 in an open conformation as in the doubly loaded Uba1-Ub(t)/Ub(a)/Cdc34 model (Fig. 5a, b, Left, Middle) and the C-terminus of Ub(t) sits in a groove surrounding the E1/E2 active sites engaging in a series of contacts our mutational analysis and previous studies suggest are important for catalysis (Fig. 3c, Supplementary Fig. 3b). Comparison of the Uba1/Cdc34-Ub(t)$^{OPEN}$ and Uba1/Cdc34-Ub(t)$^{CLOSED}$ structures reveals similar Uba1 architecture (Supplementary Fig. 7a–c, Middle, Right) but loss of aforementioned contacts between the Ub(t) C-terminus and the E1/E2 active sites and between the β-grasp domain of Ub(t) and Uba1 with concomitant formation of distinct contacts to Cdc34 (Figs. 3–5). Thus, our structural and biochemical data suggest that when E1 and E2 catalytic cysteines are brought in close proximity, a highly active Uba1 thioester linked Ub(t) tends to dissociate from Uba1 FCCH domain and associate with the crossover helix (helix α2) of the incoming Cdc34 and that contributes to E2–Ub thioester bond formation.

Collectively, the structural and biochemical data lead us to propose an affinity switch model for E1–E2–Ub thioester transfer. In the pre-transthioesterification state, potential steric clashes between Ub(t) with the incoming E2 molecule are minimized by contacts between Ub(t) and the FCCH/SCCH domains that secure Ub(t) in the open conformation. This conformation also accommodates the Ub(t) C-terminus such that key interactions with E1 and E2 that are important for catalysis can occur (Fig. 3, Supplementary Fig. 3b). As E1 and E2 active sites come into close proximity during the reaction, Ub(t) dissociates from the E1 binding site and instead engages the E2 (Fig. 5). This is significant because prior to transthioesterification the gain of contacts between Ub(t) and E2 effectively increases the affinity of E2 for the E1–Ub thioester, locking the reactants in place for catalysis, and after transthioesterification the loss of Ub(t) contacts to E1 reduces the affinity of the E2–Ub product for E1 thereby promoting product discharge and subsequent rounds of turnover. In addition to distinct contacts between Ub(t) and E2 forming, contacts between the C-terminus of Ub(t) and E1 and E2 that are important for catalysis are nearly all lost, which we speculate serves as an additional mechanism to drive the transthioesterification reaction forward by preventing the reverse reaction (i.e. reformation of the Uba1–Ub thioester bond). Interestingly, though the details differ significantly, aspects of the affinity switch model for E1–E2–Ub transthioesterification are conceptually similar to a model proposed for the Nedd8 system based on a previous structure of a doubly loaded Nedd8 E1 in complex with its E2, Ubc12 (PDB: 2NVU)[26,44].

## Discussion

The minimal mechanism for E1–E2–Ub transthioesterification involves activation of the E2 catalytic cysteine nucleophile through deprotonation of its sulfhydryl group, protonation of the leaving group (i.e. the E1 active site cysteine) after the formation of the E2–Ub thioester bond, and stabilization of the transition state that forms during catalysis. Many enzymes use amino acid side chains as general acids or bases in their catalytic mechanisms yet our structural and mutational analyses fail to identify residues that would fulfill this role in a unified mechanism for transthioesterification conserved among all 35 active human Ub E2s. The very highly conserved HPN motif of Ub E2s likely plays more of a structural than direct catalytic role[40], and acidic residues lining the channel that guides the Ub C-terminus to the E1/E2 active sites which previous studies implicated in deprotonation/pKa suppression of the incoming lysine nucleophile during isopeptide bond catalysis[41,45] do not have a profound effect on

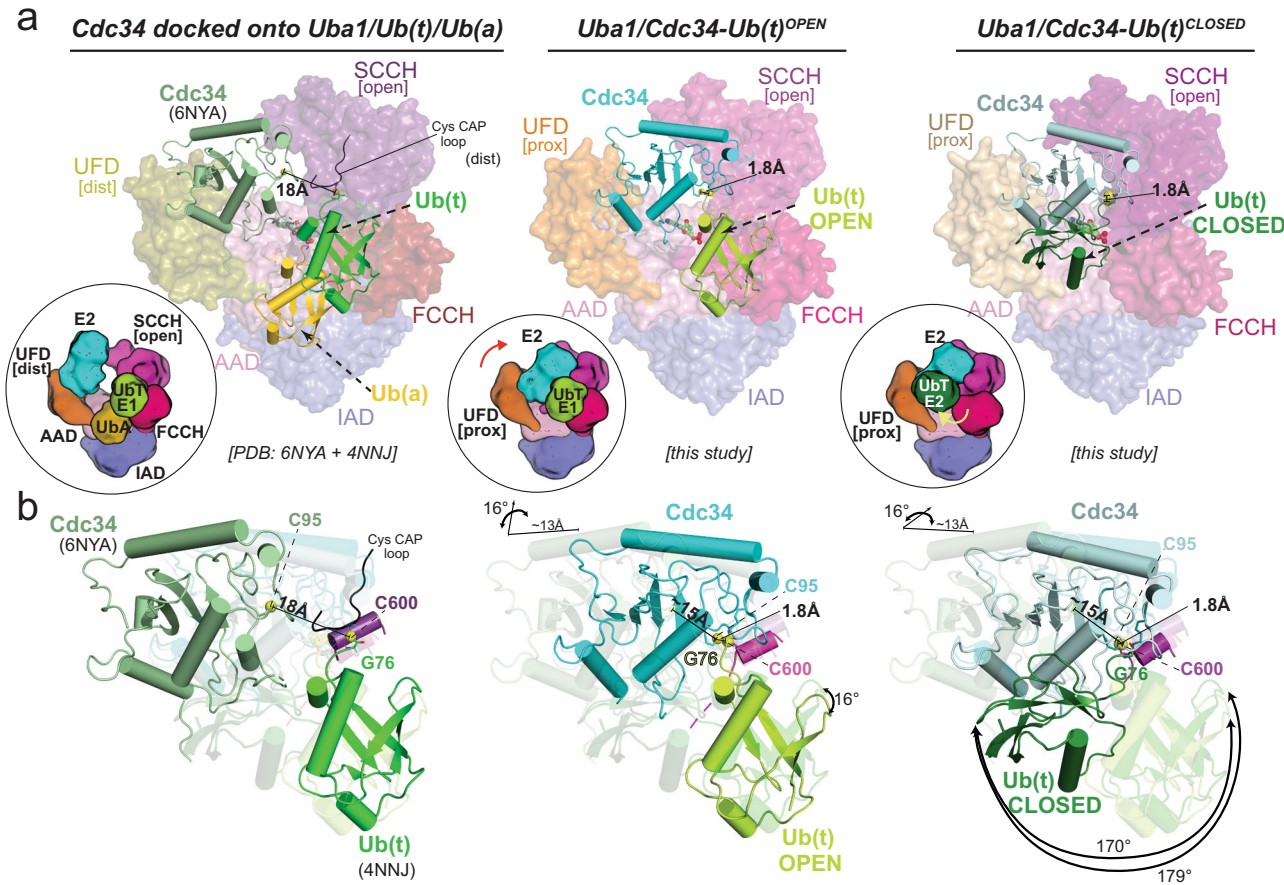

**Fig. 5 Conformational changes bring E1, E2, and Ub(t) active sites together. a** Left, Cdc34^E1-bound (PDB: 6NYA) modeled onto Uba1 doubly loaded UFD (PDB: 4NNJ) Uba1 in surface representation with domains colored and Cdc34 and Ub in cartoon representation. Sulfur atoms of catalytic cysteines are colored yellow. Middle, Uba1/Cdc34^A141K–Ub(t)^OPEN structure with Uba1 in surface representation and Cdc34 and Ub in cartoon representation. Ub(t)^OPEN sits on top of Uba1 FCCH domain. Right, Uba1/Cdc34^A141K–Ub(t)^CLOSED structure with Uba1 in surface representation and Cdc34 and Ub in cartoon representation. Ub(t)^CLOSED binds with E2 Cdc34. **b** Left, Cdc34 and Ub(t) from the docking model as shown in **a** Left, distance between E1 Cys and E2 Cys is indicated as line. Middle, Cdc34 and Ub(t) from Uba1/Cdc34^A141K–Ub(t)^OPEN structure as shown in **a** Middle. The relative distance between E2 Cys from docking model and E2 in the open structure is indicated as line. With respect to the docking model, Cdc34 rotates by 16° and translates 13 Å, while Ub(t) rotates by 16°. Right, Cdc34 and Ub(t) from Uba1/Cdc34^A141K–Ub(t)^CLOSED structure as shown in **a** Right. Cdc34 rotates by 16° and translates 13 Å with respect to the docking model, while Ub(t) rotates by 179° with respect to the docking model and rotates by 170° relative to the Ub(t)^OPEN.

transthioesterification when mutated (Fig. 3c, Fig. 4c)[16]. Altogether, this suggests crucial roles for proximity, immobilization of the E1–Ub thioester bond, and the Ub(t) affinity switch in driving transthioesterification through mechanisms proposed in this study. Lastly, the proximity of the *N*-terminus of H17 of the E1 active site to the Ub *C*-terminus suggests that positive electrostatic potential from the helix dipole of H17 could play a conserved role in catalysis by stabilization of the negative charge (Supplementary Fig. 3b, c). Interestingly, this is conceptually similar to the role of the *N*-terminus of H13 in Uba1 and H2 from the Uba2 subunit of SUMO E1, respectively, which serve as the oxyanion hole for stabilization of the transition states formed during catalysis of adenylation and thioester bond formation during Ub/Ubl activation[21,24].

Placement of our study into the broader context of previous studies on Uba1 activities allows for a more complete picture of the structural mechanisms of the enzymes of the Ub conjugation cascade to emerge, in particular, the crucially important role of conformational changes in these reactions (Fig. 6). With respect to transthioesterification, E2 is recruited to Uba1 with the UFD in the distal conformation and the E1 and E2 active sites are subsequently brought into proximity through a ~25° rotation of the UFD into proximal conformation[16,18,22] (Fig. 6). As the E1

and E2 active sites approach each other a flexible loop region that occludes the E1–Ub thioester bond becomes disordered and Ub(t) from doubly loaded Uba1 undergoes a transition from open to closed conformation in which its contacts to E1 are lost and contacts to E2 are gained which promotes E2–Ub discharge through the affinity switch mechanism (Fig. 6). During this conversion from open to closed conformation, Ub(t) contacts to Uba1 and E2 important for transthioesterification are lost which we speculate serves as an additional mechanism to drive the reaction forward by preventing the reverse reaction. Lastly, as noted above, doubly loaded Uba1 is best able to recruit an E2 and catalyze transthioesterification[14,27], and the collective data provide potential explanations for this observation. First, the Uba1 SCCH domain must adopt an open conformation in order for the E1 and E2 active sites to come into proximity during transthioesterification. The equilibrium between the open and closed states of the SCCH domain is likely shifted to the open state when the second Ub molecule is bound to the adenylation active site due to steric clashes that would occur between Ub(t) and Ub(a) when the SCCH domain approaches the closed conformation. The equilibrium between open and closed SCCH domain conformations is also shifted to the transthioesterification competent state through contacts between Ub(t) and the

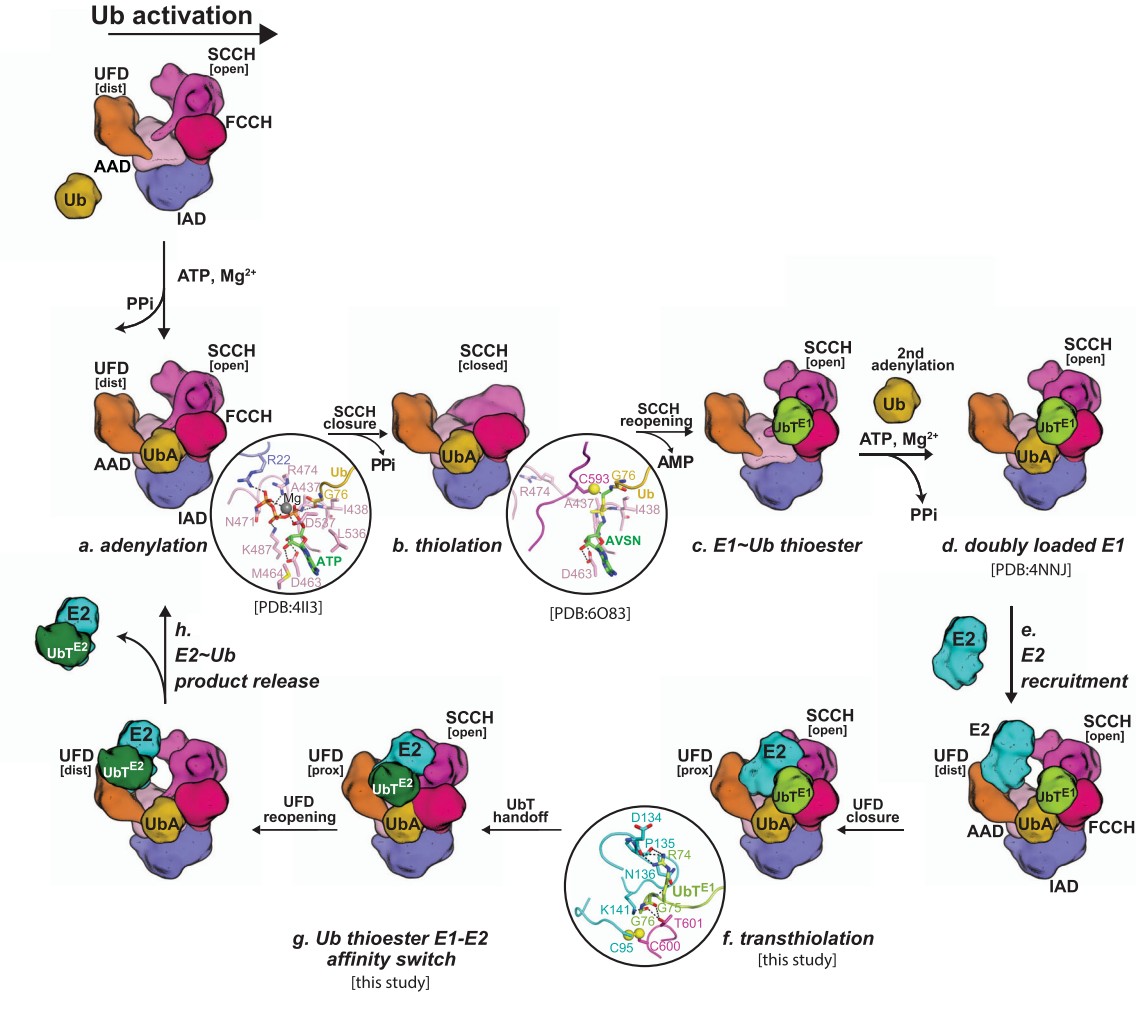

**Fig. 6 Model of E1–E2 thioester transfer reaction cycle. a** Ub E1 catalyzes adenylation of the Ub C-terminal Gly76 in the presence of ATP and Mg$^{2+}$
**b** Thiolation with E1 catalytic cysteine through disassemble of adenylation domain and a ~130° rotation of SCCH domain. **c** E1–Ub thioesterification achieved
by SCCH relocation to the original open position. **d** A second Ub binds in the adenylation active site of E1–Ub to form doubly loaded E1–Ub(t)/Ub(a). **e** A E2
is recruited to the UFD with distal configuration in doubly loaded E1–Ub(t)/Ub(a). **f** UFD rotates from distal to proximal to bring E1 E2 catalytic cysteines
into close proximity for E2–Ub thioester bond formation. **g** Ub(t) is transferred from E1 catalytic cysteine to E2 catalytic cysteine. **h** E2–Ub thioester product
is released from E1 with UFD domain in E1 rotating back to the distal conformation to continue the reaction cycle.

FCCH domain that can only occur with the SCCH domain in the open state.

There are over thirty Ub E2s in humans, which raises the question of whether the mechanistic insights into transthioesterification revealed in this study are universally applicable. Previous studies have shown that E2–Ub thioester complexes differ in their tendency to adopt the closed, open, or backbent conformations[35,36] and one of the key roles of canonical RING E3 during catalysis is to lock E2–Ub in the closed conformation that is primed for discharge of Ub onto target lysine residues[28,38]. Indeed, even E2–Ub(t)s that have a greater tendency to adopt open conformations are observed in the closed conformation in crystal structures determined both in the presence and absence of RING E3s. This suggests that many, if not most, E2s have some affinity for Ub(t) at the closed interface regardless of their preferred conformation in the absence of RING E3s (Supplementary Fig. 7d). In the context of the architecture of the complex during transthioesterification, spatial restrictions limit Ub(t) to the front surface of the E1/E2 complex, primarily due to steric clashes that would occur between E1 and Ub(t) in the backbent and many

open conformations (Supplementary Fig. 7e). We hypothesize that this spatial restriction functions to promote the closed E2–Ub(t) conformation as E2 approaches the E1 active site during transthioesterification regardless of their preferred E2–Ub (t) conformation. With that said, differences in the affinity of E2 for the Ub(t) at the closed interface may correlate with differences in the rate of product release, with higher affinity leading to higher rates of E2–Ub release and lower affinities leading to slower rates of E2–Ub release and these issues will await more detailed kinetic analyses.

Lastly, there are a total of eight E1 enzymes for Ub-like proteins, five of which share general domain organization and structural features and are thus termed canonical E1s including Uba1, Uba6, Uba7, and the heterodimeric E1s for SUMO and Nedd8 (ref. [46]). Thus, we propose that general principles of the mechanism of transthioesterification observed for Ub, including: (1) UFD rotations that place the E1 and E2 active sites in proximity to each other, (2) formation of E1–E2 interfaces that stabilize the complex in a catalytically competent conformation, and (3) conformational restriction of the E1–Ubl bond being a

key factor driving catalysis of transthioesterification are likely conserved across all canonical E1s. Domain conservation of Uba1, Uba6, and Uba7 extends to the FCCH domains, suggesting that the Ub(t) affinity switch mechanism driving the transthioesterification reaction forward may also be conserved in these systems (Supplementary Figs. 8, 9). In contrast, the FCCH domains of SUMO and Nedd8 E1s are unrelated to that of Uba1, and whether the affinity switch mechanism is conserved in these E1s and how related they might be to Uba1 is unclear and requires further studies to determine.

## Methods

**Cloning**. The DNA fragment encoding *S. cerevisiae* Uba1 residues 11-1024 (Uba1) was cloned into NcoI/XhoI sites of vector pET29NTEV with an *N*-terminal TEV-cleavable 6 × His tag[22]. The DNA fragment encoding *S. cerevisiae* Cdc34 residues 1–195 (Cdc34$^{\Delta CT}$) was cloned into NcoI/XhoI sites of vector pET29NTEV with an *N*-terminal TEV-cleavable 6 × His tag. The DNA fragment encoding *S. cerevisiae* full-length Cdc34 (residues 1–295; Cdc34$^{FL}$) was cloned as previously described[22]. *S. pombe* Uba1 (SpUba1), wild type *S. cerevisiae* Ub (Ub) and wheat ubiquitin with its seven lysines mutated to arginine (Ub$^{K7R}$) were prepared as previously described[16,18]. All point mutations were introduced using PCR-based site-directed mutagenesis. All Cdc34 mutants used in biochemical assays were introduced into Cdc34$^{FL}$. The A141K mutation of Cdc34 used to generate the Cdc34–Ub thioester mimetic for structural studies was introduced in the context of Cdc34$^{\Delta CT}$ (Cdc34$^{A141K}$). All constructs and point mutations were generated using the primer pairs described in Supplementary Table 3.

All proteins were expressed in *E. coli* BL21 (DE3) Codon Plus as previously described[47]. Large-scale cultures were grown at 37 °C until reaching the desired A600 OD, and then were placed in an ice bath for cold shock with the addition of 1.5% ethanol (v/v). After 30 minutes, the protein was induced by the addition of isopropyl-β-D-1-thioglactioside (IPTG) to a final concentration of 0.1 mM followed by shaking at 18 °C overnight. ScUba1 was grown in Terrific Broth medium as previously described[16,18], and other proteins were cultured in Luria Broth medium to A600 OD 2.0 prior to induction.

**Protein expression and purification**. Bacterial cultures of Uba1 were harvested by centrifugation and lysed by sonication in lysis buffer (20 mM Tris HCl pH 8.0, 350 mM NaCl, 20 mM Imidazole, 2 mM 2-mercaptoethanol (βme)), in the presence of DNase and Lysozyme. Cell lysate was centrifuged at $39,191 \times g$ for 30 min and the supernatant was then applied to Ni-NTA resin (QIAGEN), and the protein was eluted in buffer 20 mM Tris HCl pH 8.0, 350 mM NaCl, 250 mM Imidazole, 2 mM βme. The 6×His tag was cleaved by adding TEV protease at a ratio of 1:100 (w/w) and incubating overnight at 4 °C. After cleavage, the protein was subjected to Superdex 200 gel filtration (GE healthcare) with buffer 20 mM Tris HCl pH 8.0, 350 mM NaCl, 2 mM βme, and later the target protein was pooled and subjected to MonoQ anion exchange column (GE healthcare) with non-reducing buffer (buffer A: 20 mM Tris HCl pH 8.0, 50 mM NaCl; buffer B: 20 mM Tris HCl pH 8.0, 1000 mM NaCl) for further purification. Cdc34$^{\Delta CT}$, Cdc34$^{FL}$, and all Cdc34 mutants were purified as described for Uba1, except for using Superdex 75 gel filtration (GE healthcare) instead of Superdex 200. SpUba1 was purified as previously described[18]. Ub and Ub$^{K7R}$ were purified by Ni-NTA resin and Superdex 75 gel filtration (GE healthcare). After purification, proteins were concentrated to 5–10 mg ml$^{-1}$, aliquoted and snap frozen in liquid nitrogen.

**Cdc34$^{A141K}$–Ub thioester mimetic production and purification**. 10 μM Cdc34$^{A141K}$ was incubated with 0.08 μM Uba1, 12 μM Ub, and 0.3 mM ATP in the presence of 50 mM Tris pH 9.0, 150 mM NaCl, 5 mM MgCl$_2$, and 4 mM TCEP for 8 min at RT. The reaction mixture was subsequently placed on ice and rapidly purified over Ni-NTA resin (QIAGEN) at 4 °C. Cdc34$^{A141K}$–Ub conjugate (Cdc34$^{A141K}$–Ub(t)) was purified using Superdex 75 gel filtration (GE healthcare) followed MonoQ anion exchange (GE healthcare). The final protein was desalted into 20 mM Tris HCl pH 8.0, 50 mM NaCl, concentrated to 5 mg ml$^{-1}$ and snap frozen in liquid nitrogen.

**Uba1–Cdc34$^{A141K}$–Ub cross-linking**. Uba1/Cdc34$^{A141K}$–Ub(t) cross-linking was performed according to published methods[16,18,22]. Briefly, Cdc34$^{A141K}$–Ub(t) was treated with fresh activating buffer (20 mM Tris HCl pH 8.0, 50 mM NaCl, 2.5 mM 2,2'-Dipyridyldisulfide, 2.5% DMSO), incubated at RT for 15 min, followed by desalting into 20 mM Tris HCl pH 8.0, 50 mM NaCl, and then concentrated. Uba1 and activated Cdc34$^{A141K}$–Ub(t) were mixed at molar ratio 1:2 at RT for 1 hr. The Uba1/Cdc34$^{A141K}$–Ub(t) cross-linking mixture was further purified over a MonoQ anion exchange column (GE healthcare), concentrated to ~14.5 mg ml$^{-1}$, and snap frozen in liquid nitrogen.

**Crystallization and data collection**. Purified Uba1–Cdc34$^{A141K}$–Ub(t) crosslink complex (14.5 mg ml$^{-1}$, 100 μM) was mixed with free Ub (200 μM), 1 mM ATP

and 5 mM MgCl$_2$ prior to sparse-matrix screening in Intelli-Plate (Art Robbins Instruments) sitting drop format with 0.2 μL protein sample and 0.2 μL crystallization buffer at 18 °C. Initial crystals grew in one week. Diffraction quality crystals were grown in 0.2 M NH$_4$Ac, pH 6.91, 20% PEG3350, 0.02% (+/−)−2-Methyl-2,4-pentanediol, 0.02% 1,2,3,-Heptanetriol, 0.02% Diethylenetriaminepentakis (methylphosphonic acid), 0.02% D-Sorbitol, 0.02% Glycerol, 0.002 M HEPES sodium pH 6.8. Crystals of Uba1–Cdc34$^{A141K}$–Ub(t) grown in the presence of excess free Ub were flash-frozen in liquid nitrogen in cryoprotectant comprised of mother liquor supplemented with 1 mM ATP and 5 mM MgCl$_2$ and 25% ethylene glycol. X-ray diffraction data were collected at Advanced Photon Source (APS, Argonne, Illinois, USA), NE-CAT beamline 24-ID-C.

**Structure determination and refinement**. A data set was obtained to a resolution of 3.43 Å for crystals of Uba1–Cdc34$^{A141K}$–Ub(t) complex grown in the presence of excess Ub, Mg$^{2+}$, and ATP by merging data obtained from two crystals. The crystals are in space group P2$_1$ with unit cell dimensions $a = 95.2$ Å, $b = 272.9$ Å, and $c = 258.3$ Å and $\beta = 94.6°$. All data were indexed, integrated, and scaled using HKL2000 v717 (ref. [48]). Matthews probability calculators estimated eight copies of the complex in the asymmetric unit and the program PHASER[49] was used to find an initial molecular replacement solution using a multiple ensemble search comprising Uba1 (PDB: 3CMM [https://doi.org/10.2210/pdb3CMM/pdb]) and Cdc34$^{\Delta CT}$ (PDB: 6NYD [https://doi.org/10.2210/pdb6NYD/pdb]) as the search models. After one round of refinement the resulting maps were inspected and electron density maps validated the eight copies of both Uba1 and Cdc34$^{A141K}$ placed during molecular replacement. Further inspection revealed unambiguous electron density for eight copies of Ub(t) from the Cdc34$^{A141K}$–Ub(t) thioester mimetic into which Ub(t) was manually placed. The placement of the Cdc34 was further validated using a composite omit map. Four copies of Ub(t) were observed in the open conformation and the other four copies of Ub(t) were observed in the closed conformation as described in the results. A subsequent round of refinement validated the placement of the eight copies of Ub(t). Despite the crystals having been grown in the presence of excess Ub, Mg$^{2+}$, and ATP, electron density corresponding to Ub(a) was not observed in any of the Uba1–Cdc34$^{A141K}$–Ub(t) complexes in the asymmetric unit. Further analysis reveals that this is due to crystal packing which prevents Ub(a) from occupying its binding site on the AAD of Uba1 (Supplementary Fig. 2c). The model was refined to $R/R_{free}$ values of 0.203/0.246 via iterative rounds of refinement and rebuilding using PHENIX 1.12 (ref. [50]) and COOT 0.91 (ref. [51]). The final model contains eight copies of the Uba1–Cdc34$^{A141K}$–Ub(t) complex and seven AMP molecules. The four copies of Uba1–Cdc34–Ub(t)$^{OPEN}$ and Uba1–Cdc34–Ub(t)$^{CLOSED}$ are nearly identical and we chose the best-ordered copy for each complex (chains ABQ for open, and chains KLP for closed) to discuss throughout the manuscript and present in the figures, unless otherwise noted. A complete list of ordered/disordered residues in the 24 protein chains in the asymmetric unit is presented in Supplementary Table 2. Of note, several residues within the Uba1 crossover loop are disordered, which is likely due to the absence of Ub(a) and contacts made between the crossover loop and Ub(a). The flexible Cdc34 acidic loop is also disordered as in many other structures of E2s with acidic loops.

**E1–Ub activation and E1-E2-Ub thioester transfer assays**. Gel-based E1 thioester formation assays were performed with 5 nM E1, 5 μM Ub WT, 5 mM MgCl$_2$, 1 mM ATP, 20 mM HEPES pH7.5, 50 mM NaCl, and 0.25 mM βme for 10 s at room temperature (RT) as previously described[24]. Gel-based E1–E2–Ub thioester transfer assays were performed with 5 nM E1, 100 nM E2, 5 μM Ub, 5 mM MgCl$_2$, 1 mM ATP, 20 mM HEPES pH7.5, 50 mM NaCl and βme for 30 s at RT as previously described[16]. Reactions were initiated by adding ATP and were terminated by adding non-reducing Urea SDS-PAGE buffer and subjected to SDS-PAGE, 180 V constant for 35 min. The gels were stained with Sypro Ruby (Biorad) and visualized with a ChemiDoc MP (Biorad). Data quantification was conducted using densitometry in ImageJ 1.53 software and analyzed using Prism 7.0a (GraphPad). Densitometry measurements were normalized as a percentage of the control WT assay on the same gel. Data are represented as an average of three technical replicates with ± standard deviation error bars. Unprocessed images of representative gels for all biochemical assays are provided in the Source Data file.

**Uba1/Cdc34$^{A141K}$–Ub(t) and Ub(a) co-migration analysis**. Purified Uba1$^{WT}$–Cdc34$^{A141K}$–Ub(t) complex was mixed with a 2-fold molar excess of Ub in the presence of 1 mM ATP and 5 mM MgCl$_2$ and incubated on ice for 1 h. As one negative control, purified Uba1$^{WT}$–Cdc34$^{A141K}$–Ub(t) complex was mixed with a 2-fold molar excess of Ub in the presence of 5 mM MgCl$_2$ but without ATP and incubated on ice for 1 h. As another negative control, purified adenylation deficient mutation Uba1$^{D544A}$–Cdc34$^{A141K}$–Ub(t) complex was mixed with a 2-fold molar excess of Ub in the presence of 1 mM ATP and 5 mM MgCl$_2$ and incubated on ice for 1 h. The mixtures were then separately applied to a Superose 12 10/300 GL size-exclusion chromatography column equilibrated with 20 mM Tris HCl pH 8.0, 50 mM NaCl, 1 mM ATP and 5 mM MgCl$_2$. Fractions were subjected to SDS-PAGE and stained with Coomassie Blue. Uba1$^{WT}$–Cdc34$^{A141K}$–Ub(t), Uba1$^{D544A}$–Cdc34$^{A141K}$–Ub(t) and free Ub were individually analyzed as controls under the same conditions.

**Thermal shift assay**. 5 μg Cdc34 Wild type or mutants proteins was mixed with 2.5× SYPRO Orange dye (Thermo Fisher) in 20 mM HEPES pH 7.5, 100 mM NaCl to 20 μL in MicroAmp Fast optical 96-well reaction plate (Life Technologies). Each sample was prepared in triplicate. The 96-well was sealed with MicroAmp Optical Adhesive Film (LifeTechnologies) and then was placed into QuantStudio 3 qRT-PCR (Appliedbiosystems). Running the melt curve method: selecting continuous collection; 25 °C 2 min, 1.6 °C /s; 0.05 °C/s ramp; 95 °C 2 min. Data collection was saved for constructing melt curves and determining melting temperature by Prism 7.0a (GraphPad).

**Reporting summary**. Further information on research design is available in the Nature Research Reporting Summary linked to this article.

## Data availability

Atomic coordinates and structure factors are deposited in the RCSB with accession code 7K5J. Any additional data may be requested from the corresponding author. The structure data used from RCSB are listed below: PDB: 4NNJ, PDB: 6NYA, PDB: 2NVU, PDB: 3CMM, PDB: 6NYD, PDB: 4MDK, PDB: 1Z5S, PDB: 4AUC, PDB: 4AP4, PDB: 4P5O, PDB: 4II3, PDB: 6O83, PDB: 5IFR, PDB: 3A33, PDB: 3JW0, PDB: 5KNL, PDB: 6D68, PDB: 4JQU, PDB: 4LAD, PDB: 6OP8, PDB: 3K9O, PDB: 1FTX, PDB: 6CYO, PDB: 1JAS, PDB: 4R62, PDB: 5EGG, PDB: 1YRV, PDB: 3BZH, PDB: 1Y6L, PDB: 4II2, PDB: 1QCQ, PDB: 2F4W, PDB: 2Z5D, PDB: 3RCZ, PDB: 2EKE, PDB: 3FN1, PDB: 3O2U, PDB: 6S53, PDB: 5NGZ, PDB: 1JAT, PDB: 1ZDN, PDB: 4BWF, PDB: 4YII, PDB: 2MT6, PDB: 5A4P, PDB: 3ZEG, PDB: 4Q5E, PDB: 1WZV, PDB: 2QGX, PDB: 1ZUO, PDB: 5TUT, PDB: 2KJH, PDB: 5DFL, PDB: 5BNB, PDB: 5ULF, PDB: 6DC6, PDB: 1Z7L, PDB: 3KYC, PDB: 1R4M, PDB: 3RZ3, PDB: 1TTE, PDB: 4ONM. Source data are provided with this paper.

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

## Acknowledgements
The authors thank Adam J. Smolka, Kristin E. Cano, and Alex B. Taylor for critically reading the manuscript and Katelyn Williams and Liping Liu for technical support. The X-ray diffraction data were collected at SER-CAT 22-ID and NE-CAT 24-ID-E beamlines at the Advanced Photon Source, Argonne National Laboratory. This work is based upon research conducted at the Northeastern Collaborative Access Team beamlines, which are funded by the National Institute of General Medical Sciences from the National Institutes of Health (P30 GM124165). The Eiger 16M detector on 24-ID-E is funded by a NIH-ORIP HEI grant (S10OD021527). This research used resources of the Advanced Photon Source, a U.S. Department of Energy (DOE) Office of Science User Facility operated for the DOE Office of Science by Argonne National Laboratory under Contract No. DE-AC02-06CH11357. Research reported in this publication was supported by the NIH R01 GM115568, R01 GM128731 and CPRIT RR200030 (S.K.O.). Z.L. was partially supported by a Hollings Cancer Center Postdoctoral Fellowship. This work is based upon research conducted in the X-ray Crystallography Core Laboratory, a part of the Institutional Research Cores at the University of Texas Health Science Center at San Antonio supported by the Office of the Vice President for Research and the Mays Cancer Center Drug Discovery and Structural Biology Shared Resource (NIH P30 CA054174). The content of this study is solely the responsibility of the authors and does not necessarily represent the official views of the NIH.

## Author contributions
Structural experiments and analysis were performed by L.Y., Z.L., and S.K.O. L.Y., and Z.L., and M.J.A. conducted biochemical assays. The manuscript was written by L.Y. and S.K.O.

## Competing interests
The authors declare no competing interests.
