## [Peer Review File · Nature Communications]

REVIEWER COMMENTS

Reviewer #1 (Remarks to the Author):

The manuscript "Structure of an E1-E2-ubiquitin thioester mimetic: insights into mechanisms of transthioesterification" by Yuan et al. reports the crystal structure of Cdc34 that is covalently linked to Uba1 and ubiquitin via a disulfide bond and an isopeptide bond, respectively. The structure reveals two conformations for this complex, which the authors suggest reflect the pre and post Ub transfer. Specifically, it shows two different conformations of the Ub(T) that occurs during Ub transfer. This work therefore advances our understanding of how Ub is transferred from E1 to E2 and provides novel structural insights on this process.

The manuscript is well written and nicely explains how this work contributes to our understanding of transthiolation.

I have a few minor points.

To support their sizing experiments of a complex possessing both Ub(a) and Ub(t) the authors can use SEC-MALS to show the molecular weight of the complex in solution. Also, why does Ub run above the 15kd marker in sup fig 1, while in the other gels it seems to be below this marker?

In fig 3c the mutations in Cdc34 show significant defects in transfer. However, one can argue that these are structural mutations that affect the stability of the protein. The authors should address this point using simple thermal stability assays.

Regarding the mutations in the E1- they hardly show defects in transfer. Can the authors show that combinations of these mutations indeed show defects in transfer, or alternatively, perform mutagenesis not only to Ala.

please describe how the docking of Cdc34 to the structure of doubly loaded Uba1 was done.

Fig3d - Do the pink shaded boxes show residues that interact with Ub(t) ? if this is so, then what does “ contacts to Uba1 FCCH mean?

Fig 4d - what is the difference between the blue and yellow boxes? it says “contacts of Ub” for both

Reviewer #2 (Remarks to the Author):

In this manuscript, Yuan et al. present a crystal structure of a covalently trapped E1-E2-ubiquitin thioester mimetic that provides snapshots of ubiquitin transfer from E1 to E2. While existing structures of E1-ubiquitin thioester complex and E1-E2 complexes covalently linked via their cysteines provided insights into the mechanism of E1-E2 transthiolation reaction, these structures did not fully capture the transition state intermediate of the reaction and therefore lack detailed reaction mechanism. To capture this transition state, the authors used a previously described strategy for the SUMO system to link ubiquitin to E2 Cdc34 and subsequently covalently link this Cdc34's catalytic cysteine to E1's catalytic cysteine to obtain a trapped E1-E2-ubiquitin thioester mimetic. This trapped complex retains adenylation activity suggesting the complex is functional. In the crystal lattice, this trapped complex adopts two configurations where ubiquitin thioester (Ub(t)) exists in an open and a closed conformations. The open Ub(t) contacts E1's FCCH and SCCH, whereas the closed Ub(t) packs against Cdc34 and did not contact E1. Biochemical analyses showed that mutations on E1's FCCH and SCCH regions that contact open Ub(t) had subtle effect on the E1-E2 reaction. In contrast, mutations on Cdc34 surfaces that contact closed Ub(t) had more profound defect in the E1-E2 reaction. Based on these findings, the authors posit an affinity switch model. In this model, E1-ubiquitin thioester in an open Ub(t) conformation recruits E2 via the UFD. UFD rotation enables juxtaposition the catalytic cysteines of E1 and E2 and the high affinity binding between E2 and closed Ub(t) drives the reaction forward.

This is an excellent work that provides new insights into E1-E2 reaction mechanism. It will be of interest to researchers working in the ubiquitin related field and protein structure and mechanism. I only have a few comments.

1. Is the affinity switch model applicable to other E2s? Olsen's group has previously shown that Cdc34-ubiquitin complex prefers to form the closed conformation via the alpha 2 helix (Williams et al. 2019 Nature Communication). The closed Ub(t) conformation observed here is very similar to their prior structure work on Cdc34-ubiquitin complex. This suggests that Cdc34 has a preference to form the closed conformation. Since closed Ub(t) only contacts Cdc34, the affinity switch model

seemingly depends on the ability of E2 to form the closed Ub(t) conformation. Solution analyses of several E2-ubiquitin complexes showed that they populate in a more open conformation. It would be worthwhile to discuss the relevance of this model in the light of other E2s.

2. The main text indicated that the crystal packing prevents Ub(a) from occupying the binding site on the AAD. It was not clear from Supplementary Figure 1c that a symmetry-related molecule occlude Ub(a) binding site. The dark pink structure indicated in the panel corresponds to the FCCH and the insets appear to show that FCCH partially occludes Ub(a) binding from structure modelling. Did the crystal packing perturb FCCH orientation and consequently occludes Ub(a) from occupying the AAD binding site? It would be best to provide explanation in the Figure legend.

3. The figures/supplementary figures do not appear in sequential order in the main text. Please check throughout.

Reviewer #3 (Remarks to the Author):

Review of 'Structure of an E1-E2-ubiquitin thioester mimetic: insights into mechanisms of transthioesterification'

The authors report the structure and functional characterization of a covalently trapped E1-E2-ubiquitin thioester complex.

Formation of a transient and low affinity E1-E2-Ubiquitin thioester (Ub(t)) complex required two distinct covalent modifications.

First, a E2-thioester mimic involved mutation of Ala141Lys near the catalytic cysteine that was subsequently linked to Ub via an isopeptide bond. This species is resistant to the presence of DTT.

The E1-E2 interface was then stabilized by disulfide bond formation between their respective catalytic cysteines.

The functionality of the complex was confirmed by free Ubiquitin binding using SEC.

The resultant stabilized E1-E2-Ub(T) complex was crystallized and the structure was determined at 3.3 Å resolution.

Ub(t) (Ub thioester) was visualized in two distinct conformations within the 8 copies of the complex in the asymmetric unit of the crystal.

4 copies of the complex appear with Ub(t) in an open conformation (relative to previously reported E2~Ub structures) making contacts largely with the E1. This conformation is proposed to represent a snapshot of E1/Ub/E2 complex pre-thioester transfer of Ub from the E1 to the E2.

4 copies of the complex appear with Ub(t) in a closed conformation (relative to previously reported E2~Ub structures) making extensive contacts with the E2. This is proposed to represent a snapshot of E1/Ub/E2 complex post-thioester transfer of Ub from the E1 to the E2.

The authors rationalize that both open and closed structures have relevance for E1 to E2 thioester transfer of Ub with a defined temporal order (open precedes closed).

To validate the structures, a series of single point mutations on the E1/Ub(t) open and E2/Ub(t) closed interfaces were generated and tested for formation of E2~Ub.

Mutations on the contact surface between the E1 and Ub(t) in the open configuration were generally tolerated (50-75% activity of E1-wt).

Mutations on the contact surface between E2 and Ub(t) in the open configuration compromised activity (~25% activity of E2-wt)

Mutations on the contact surface between E2 and Ub(t) in the closed configuration compromised activity (~25% activity of E2-wt) (Fig.3C, Fig.4C).

(note that the contact surfaces between Ub(t) and E2 in the open and closed configurations are partially overlapping which would make a clear separation of function difficult).

Based on the presented results, the authors propose a model in which transfer of Ub from the E1 is enhanced by a Ub-mediated affinity switch that drives the reaction forward (Fig.6).

Overall, the authors present important snapshots of a complicated and important reaction mechanism. The analysis of structures is very detailed and thoughtful. For this reason alone, the presented work is sufficient to warrant publication in a well respected journal such as Nature Comm.

My only concern is that while the authors provide a compelling rationalization of the affinity switch /temporal reaction mechanism, the mutagenesis data do not unequivocally validate their model.

In fact, the data are just as consistent with a reverse version of their temporal model. For this reason, the authors should either acknowledge the limitations of their structure interpretations (for example don't state that the model is proven in the discussion) or provide additional experimental validation of their model.

For example, can the authors use kinetic analyses of Ub thioester transfer to nail down the temporal characteristics of their model? In the simplest sense, the authors model predicts that mutation of the open conformation Ub contact surface would be most directly detrimental to the transfer of Ub from E1 to E2, whereas mutation of the closed conformation Ub contact surface would be more detrimental to substrate release following Ub transfer.

The authors should also consider addressing the following minor points.

Comments:

1) Presumably the high Rmerge value (0.287 overall) of the diffraction data was due to merging of two crystal datasets? This reviewer is curious why two crystals needed to be merged.

2) In Fig 1B, formation of the higher molecular weight Cdc34-A141K~Ub conjugate was shown to be resistant to the presence of DTT. In Fig 3C and Fig 4C the authors should similarly show that the

E2~Ub intermediates are DTT-sensitive thioester to confirm that an E2~Ub charging complex was indeed being assessed. Presumably E2~Ub formation assays in Fig 3C and 4C were conducted with Cdc34-wt and not the Cdc34-A141K mutant.

3) Density for the E1/E2 Cys-Cys disulfide looks continuous and agrees with the structural model in Supp. Fig 2C. However, the E2-Ub isopeptide bond with E2-Lys141 does not appear well ordered in the view shown in Supp. Fig 2B. How did the authors confirm the linkage between Ub and E2?

4) One important missing control for the E1 mutational analysis is to show that the E1 enzyme is not impaired for charging by Ubiquitin. Provide quantification of E1~Ub formation in Supp. Fig 3a.

5) Do the 4 open state structures display similar crystal packing arrangements in the unit cell that could account for the positioning of Ub(t)? Do the 4 closed state structures display similar packing arrangements in the unit cell that could account for the positioning of Ub(t)?

If the packing arrangements for all 8 molecules in the asymmetric unit are very different, and yet only two conformations of Ub were observed, then that would provide further support to the notion that the conformations are more likely functionally relevant rather than coincidental crystal packing artifacts.

REVIEWER COMMENTS:

Reviewer #1 (Remarks to the Author):

The manuscript " Structure of an E1-E2-ubiquitin thioester mimetic: insights into mechanisms of transthioesterification" by Yuan et al. reports the crystal structure of Cdc34 that is covalently linked to Uba1 and ubiquitin via a disulfide bond and an isopeptide bond, respectively. The structure reveals two conformations for this complex, which the authors suggest reflect the pre and post Ub transfer. Specifically, it shows two different conformations of the Ub(T) that occurs during Ub transfer. This work therefore advances our understanding of how Ub is transferred from E1 to E2 and provides novel structural insights on this process.

The manuscript is well written and nicely explains how this work contributes to our understanding of transthiolation.

We are grateful to the reviewer for their positive comments and thoughtful analysis of our study.

I have a few minor points.

To support their sizing experiments of a complex possessing both Ub(a) and Ub(t) the authors can use SEC-MALS to show the molecular weight of the complex in solution.

We appreciate the reviewer's concern and made further efforts to confirm that we have indeed isolated a covalently trapped, doubly loaded Uba1-Cdc34-Ub(a)-Ub(t) complex. To address this issue we took advantage of the fact that Ub(a) but not free Ub co-migrates with Uba1 in analytical gel filtration experiments due to the significantly higher Uba1 binding affinity of Ub(a) compared to free Ub. Based on this, we have performed the following two additional analytical gel filtration experiments which are now included in Supplementary Information:

First, as shown in new Supplementary Fig. 1b, our covalently trapped Uba1-Cdc34-Ub(t) complex comigrates with Ub only in the presence of ATP•Mg²⁺ (left vs center panels). Second, Uba1-Cdc34-Ub(t) complex harboring a Uba1 D544A mutation that abolishes its adenylation activity is unable to comigrate with Ub despite the presence of ATP•Mg²⁺ (right panel). This data further reinforces our conclusion that the covalently trapped Uba1-Cdc34-Ub(t) complex retains key structural features required for Ub adenylation, and that Ub only comigrates with Uba1-Cdc34-Ub(t) presumably due to the significantly lower Uba1 binding affinity of free Ub compared to Ub that has been adenylated by Uba1. Altogether, the data strongly suggest that the sample we subjected to crystallization experiments harbored both Ub(a) and Ub(t) as the complex was reconstituted with wildtype Uba1 and in the presence of ATP•Mg²⁺.

SEC-MALS is an excellent technique to address such an issue but we were unable to secure access to a suitable instrument within the time frame of revising the manuscript. We also had some concerns about the ability to differentiate between Uba1-Cdc34-Ub(t) (~162 kDa) and Uba1-Cdc34-Ub(a)-Ub(t) (~170 kDa) due to the relatively minor ~8 kDa difference in molecular weight between the complexes.

Also, why does Ub run above the 15kd marker in sup fig 1, while in the other gels it seems to be below this marker?

We thank the reviewer for pointing this out. Ub consistently runs below the 15kDa marker in all of the gels presented in the manuscript. Our labels for the 10 and 15 kDa molecular weight standards were poorly placed in Supplementary Fig.1b and we have addressed this in the revised manuscript. We apologize for the confusion.

In fig 3c the mutations in Cdc34 show significant defects in transfer. However, one can argue that these are structural mutations that affect the stability of the protein. The authors should address this point using simple thermal stability assays.

We appreciate the reviewer's suggestion, and have performed the recommended thermal shift assays, which are presented in new Supplementary Fig, 5d. The results show that all of the Cdc34 mutants, with the exception of N87A, have similar melting temperatures (51.5 ~ 54.6 °C) as the wildtype (52.9 °C), which suggests the mutations do not affect the stability of the proteins. Regarding the N87A mutant, as noted in the manuscript, Asn87 resides within the very highly conserved 'HPN' motif of E2s and participates in a network of intramolecular hydrogen bonds with residues from the α2-α3 loop of Cdc34 (Asn136 and Ser139; new Supplementary Fig. 3b and 5c Right panels) and these interactions stabilize the active site for catalysis, as previously suggested (Dou et. al, Nat Struct Mol Biol, 2012; Berndsen et al., Nat Chem Biol, 2013). The lower melting temperature (relative to wild type) we observe for the Cdc34 N87A mutant suggests that its significantly diminished transthioesterification activity is at least partially due to structural effects. We have revised the manuscript to incorporate the above data and conclusions.

Regarding the mutations in the E1- they hardly show defects in transfer. Can the authors show that combinations of these mutations indeed show defects in transfer, or alternatively, perform mutagenesis not only to Ala.

We thank the reviewer for their comments and have performed a series of experiments based on their suggestions. First, we generated Uba1 variants harboring multiple mutations of residues at the Uba1^{FCCH}/Ub(t) interface including D188L/E190L, D188L/T196V, E190L/T196V, D188L/E190L/T196V and D188L/E190L/T196V/L198R. We also generated Uba1 variants harboring multiple mutations of residues at the Uba1^{SCCH}/Ub(t) interface such as N703D/E705K, and T601V/N703D/E705K. Control E1~Ub thioester formation assays showed the seven E1 combination mutations have no affect on Ub activation. In E1-E2 transtioesterification assays, we observed a range of activities for the seven E1 combination mutants with the most pronounced defects being observed for the D188L/E190L/T196V triple mutant at the Uba1 FCCH/Ub(t) interface (~60% activity relative to WT), and the T601V/N703D/E705K triple mutant at the Uba1 SCCH/Ub(t) interface (~30% activity relative to WT). These results are consistent with this interface being transient and low affinity and requiring more than conservative alanine substitutions to disrupt to the extent that diminished transtioesterification activities are observed. The data is presented in new Supplementary Figs. 4a and b and we have edited the text to include description of the data and interpretations.

Please describe how the docking of Cdc34 to the structure of doubly loaded Uba1 was done.

During initial structure determination using molecular replacement (MR), we took two different approaches to placing Cdc34 into the electron density maps. First, we solved the structure using only Uba1 as a search model and noted that after a single round of refinement that there was very strong FoFc electron density where E2 would be located based on previously determined E1/E2 structures for all eight copies of Uba1 that were placed. In parallel, we used PHASER to perform MR using a multi-ensemble search in which Uba1 was the first search model followed by Cdc34. This resulted in eight copies of Uba1 being placed followed by eight copies of Cdc34. TFZ scores of the Uba1/Cdc34 solution were significantly higher than for Uba1 alone and the resulting electron density maps for all eight copies of Cdc34 were unambiguous, including in composite omit maps. Further analysis showed that Cdc34 from the resulting Uba1-Cdc34 complex corresponded to the FoFc electron density noted above for first approach in which MR and refinement were performed in the absence of Cdc34. As a result, we used the Uba1-Cdc34 solution obtained through the multi-ensemble MR approach for subsequent placement of Ub(t) and downstream model building and refinement as described in the Methods section.

Fig3d - Do the pink shaded boxes show residues that interact with Ub(t) ? if this is so, then what does “ contacts to Uba1 FCCH mean?

Yes they do, thank you for pointing out this confusing labeling. We have reworded the labels in the revised manuscript to make this clearer.

Fig 4d - what is the deference between the blue and yellow boxes? it says “contacts of Ub” for both

We apologize for the confusion, which is the result of unclear labeling. Blue and yellow boxes are used to differentiate the E2-Ub(t) interactions observed in this study (blue) from previously reported E2-Ub(t), E2-SUMO(t), and E2-Nedd8(t) structures (yellow). We have updated the labels to make this clearer and thank the reviewer for pointing this out.

Reviewer #2 (Remarks to the Author):

In this manuscript, Yuan et al. present a crystal structure of a covalently trapped E1-E2-ubiquitin thioester mimetic that provides snapshots of ubiquitin transfer from E1 to E2. While existing structures of E1-ubiquitin thioester complex and E1-E2 complexes covalently linked via their cysteines provided insights into the mechanism of E1-E2 transthiolation reaction, these structures did not fully capture the transition state intermediate of the reaction and therefore lack detailed reaction mechanism. To capture this transition state, the authors used a previously described strategy for the SUMO system to link ubiquitin to E2 Cdc34 and subsequently covalently link this Cdc34's catalytic cysteine to E1's catalytic cysteine to obtain a trapped E1-E2-ubiquitin thioester mimetic. This trapped complex retains adenylation activity suggesting the complex is functional. In the crystal lattice, this trapped complex adopts two configurations where ubiquitin thioester (Ub(t)) exists in an open and a closed conformations. The open Ub(t) contacts E1's FCCH and SCCH, whereas the closed Ub(t) packs against Cdc34 and did not contact E1. Biochemical analyses showed that mutations on E1's FCCH and SCCH regions that contact open Ub(t) had subtle effect on the E1-E2 reaction. In contrast, mutations on Cdc34 surfaces that contact closed Ub(t) had more profound defect in the E1-E2 reaction. Based on these findings, the authors posit an affinity switch model. In this model, E1-ubiquitin thioester in an open Ub(t) conformation recruits E2 via the UFD. UFD rotation enables juxtaposition the catalytic cysteines of E1 and E2 and the high affinity binding between E2 and closed Ub(t) drives the reaction forward.

This is an excellent work that provides new insights into E1-E2 reaction mechanism. It will be of interest to researchers working in the ubiquitin related field and protein structure and mechanism. I only have a few comments.

Thank you for the positive comments and thoughtful analysis of our study.

1. Is the affinity switch model applicable to other E2s? Olsen's group has previously shown that Cdc34-ubiquitin complex prefers to form the closed conformation via the alpha 2 helix (Williams et al. 2019 Nature Communication). The closed Ub(t) conformation observed here is very similar to their prior structure work on Cdc34-ubiquitin complex. This suggests that Cdc34 has a preference to form the closed conformation. Since closed Ub(t) only contacts Cdc34, the affinity switch model seemingly depends on the ability of E2 to form the closed Ub(t) conformation. Solution analyses of several E2-ubiquitin complexes showed that they populate in a more open conformation. It would be worthwhile to discuss the relevance of this model in the light of other E2s.

We thank the reviewer for these very insightful comments and suggestions. Indeed, different E2~Ub thioester complexes have been observed to differ in their tendency to adopt the closed, open, or backbent conformation (Pruneda et al, Biochemistry 2011; Page et al. Biochemistry 2012). Current dogma for E2s that function with RING E3s is that the active conformation of E2~Ub complexes is the 'closed' conformation which is stabilized by contacts between Ub(t) and RING E3. Indeed, even E2~Ub(t)s that the aforementioned studies indicate have a greater tendency to adopt open conformations are observed in the closed conformation in crystal structures determined both in the presence and absence of RING E3s. This suggests that many, if not most, E2s have some affinity for Ub(t) at the closed interface regardless of their preferred conformation in the absence of RING E3s. In the context of E1 and transthioesterification, spatial restrictions limit Ub(t) to the 'front' surface of the E1/E2 complex, primarily due to steric clashes that would occur with Ub(t) in the backbent and many open conformations that have been observed in previous structures. We hypothesize that this spatial restriction functions akin to RING E3 to promote the closed E2~Ub(t) conformation as E2 approaches the E1 active site during transthioesterification, primarily due to proximity effects. With respect to different E2s, we speculate that the affinity switch model could be applicable to E2s regardless of their preferred E2~Ub(t) conformation. With that said, differences in the affinity of E2 for the Ub(t) at the closed interface may correlate with differences in the rate of product release, with higher affinity leading to higher rates of E2~Ub release and lower affinities leading to slower rates of E2~Ub release. As noted below, we are planning on performing more detailed kinetic analyses to address these issues along with the contributions of the closed and open Ub(t) interfaces on these processes. We have included this in the Discussion of the revised manuscript.

2. The main text indicated that the crystal packing prevents Ub(a) from occupying the binding site on the AAD. It was not clear from Supplementary Figure 1c that a symmetry-related molecule occlude Ub(a) binding site. The dark pink structure indicated in the panel corresponds to the FCCH and the insets appear to show that FCCH partially occludes Ub(a) binding from structure modelling. Did the crystal packing perturb FCCH orientation and consequently occludes Ub(a) from occupying the AAD binding site? It would be best to provide explanation in the Figure legend.

Yes, crystal packing indeed perturbs the orientation of the FCCH domain such that the Ub(a) binding site on the AAD is partially occluded. We believe that this is why Ub(a) was not observed in the structure despite being present in the sample that was subjected to crystallization (please see response to reviewer 1 above). We appreciate the reviewer's suggestion, and have added a more detailed explanation in the figure legend for Supplementary Fig. 2c.

3. The figures/supplementary figures do not appear in sequential order in the main text. Please check throughout.

Thank you for pointing this out, we have fixed this inconsistency.

Reviewer #3 (Remarks to the Author):

Review of 'Structure of an E1-E2-ubiquitin thioester mimetic: insights into mechanisms of transthioesterification'

The authors report the structure and functional characterization of a covalently trapped E1-E2-ubiquitin thioester complex.

Formation of a transient and low affinity E1-E2-Ubiquitin thioester (Ub(t)) complex required two distinct covalent modifications.

First, a E2-thioester mimic involved mutation of Ala141Lys near the catalytic cysteine that was subsequently linked to Ub via an isopeptide bond. This species is resistant to the presence of DTT.

The E1-E2 interface was then stabilized by disulfide bond formation between their respective catalytic cysteines.

The functionality of the complex was confirmed by free Ubiquitin binding using SEC.

The resultant stabilized E1-E2-Ub(T) complex was crystallized and the structure was determined at 3.3 Å resolution.

Ub(t) (Ub thioester) was visualized in two distinct conformations within the 8 copies of the complex in the asymmetric unit of the crystal.

4 copies of the complex appear with Ub(t) in an open conformation (relative to previously reported E2~Ub structures) making contacts largely with the E1. This conformation is proposed to represent a snapshot of E1/Ub/E2 complex pre-thioester transfer of Ub from the E1 to the E2.

4 copies of the complex appear with Ub(t) in a closed conformation (relative to previously reported E2~Ub structures) making extensive contacts with the E2. This is proposed to represent a snapshot of E1/Ub/E2 complex post-thioester transfer of Ub from the E1 to the E2.

The authors rationalize that both open and closed structures have relevance for E1 to E2 thioester transfer of Ub with a defined temporal order (open precedes closed).

To validate the structures, a series of single point mutations on the E1/Ub(t) open and E2/Ub(t) closed interfaces were generated and tested for formation of E2~Ub.

Mutations on the contact surface between the E1 and Ub(t) in the open configuration were generally tolerated (50-75% activity of E1-wt).

Mutations on the contact surface between E2 and Ub(t) in the open configuration compromised activity (~25% activity of E2-wt)

Mutations on the contact surface between E2 and Ub(t) in the closed configuration compromised activity (~25% activity of E2-wt) (Fig.3C, Fig.4C).

(note that the contact surfaces between Ub(t) and E2 in the open and closed configurations are partially overlapping which would make a clear separation of function difficult).

Based on the presented results, the authors propose a model in which transfer of Ub from the E1 is enhanced by a Ub-mediated affinity switch that drives the reaction forward (Fig.6).

Overall, the authors present important snapshots of a complicated and important reaction mechanism. The analysis of structures is very detailed and thoughtful. For this reason alone, the presented work is sufficient to warrant publication in a well respected journal such as Nature Comm.

We are grateful to the reviewer for the thorough analysis of our manuscript and for the positive comments.

My only concern is that while the authors provide a compelling rationalization of the affinity switch /temporal reaction mechanism, the mutagenesis data do not unequivocally validate their model.

In fact, the data are just as consistent with a reverse version of their temporal model. For this reason, the authors should either acknowledge the limitations of their structure interpretations (for example don't state that the model is proven in the discussion) or provide additional experimental validation of their model.

We agree that there are limitations of our interpretations of our structural and biochemical data. As suggested, we have more prominently acknowledged these limitations and alternative interpretations in the results, and in particular, in the discussion. We plan on pursuing other aspects of these issues in future studies, including investigations on other E2s.

For example, can the authors use kinetic analyses of Ub thioester transfer to nail down the temporal characteristics of their model? In the simplest sense, the authors model predicts that mutation of the open conformation Ub contact surface would be most directly detrimental to the transfer of Ub from E1 to E2, whereas mutation of the closed conformation Ub contact surface would be more detrimental to substrate release following Ub transfer.

We thank the reviewer for their suggestions and this is an excellent point regarding distinctions in how open and closed Ub(t) interactions are predicted to affect transfer of Ub from E1 to E2 versus substrate release. We have mentioned this in the revised manuscript. As noted above, we do intend to investigate these issues in future studies, and in particular, we are designing kinetic experiments that will require access to a quench flow instrument and are in the process of locating an instrument we can use for these experiments. This will take longer to complete and so we feel these experiments are beyond the scope of this study.

The authors should also consider addressing the following minor points.

Comments:

1) Presumably the high Rmerge value (0.287 overall) of the diffraction data was due to merging of two crystal datasets? This reviewer is curious why two crystals needed to be merged.

Yes, the high Rmerge value is due to merging of two crystal datasets. Although the Rmerge values of data collected from single crystals were superior to the merged data and all of the overall features of our structure including 4 copies off the complex each in the open and closed conformations, the electron density maps for the merged data were clearly superior overall, particularly in some key parts of the structure. For example, electron density for parts of the E1/E2 active sites and in particular the isopeptide bond linking Cdc34 to Ub(t) is significantly stronger for maps derived from the merged data and this was the primary reason we chose this data for model building and refinement. We speculate that this may be due to the higher redundancy and/or I/σ of the merged data. Lastly, while the Rmerge is indeed high for the merged data sets, we would like to note that the high redundancy is likely a contributing factor to this in addition to merging, as the R_pim value (which is less affected by high redundancy) for the merged data is 0.084.

2) In Fig 1B, formation of the higher molecular weight Cdc34-A141K~Ub conjugate was shown to be resistant to the presence of DTT. In Fig 3C and Fig 4C the authors should similarly show that the E2~Ub intermediates are DTT-sensitive thioester to confirm that an E2~Ub charging complex was indeed being assessed. Presumably E2~Ub formation assays in Fig 3C and 4C were conducted with Cdc34-wt and not the Cdc34-A141K mutant.

We understand the reviewer's concern and note that the E2~Ub formations assays in Fig 3C and 4C were conducted with wild type Cdc34 (not A141K mutant). We have clarified this in the revised manuscript. We also performed the suggested experiments for the E1 and E2 mutants studied in Fig. 3c and Fig. 4c and the results are presented in Supplementary Fig. 4a and Supplementary Fig. 5c. The results show that all of the E1~Ub and E2~Ub intermediates observed in these experiments are DTT-sensitive thioester products rather than DTT-resistant isopeptide linkages.

3) Density for the E1/E2 Cys-Cys disulfide looks continuous and agrees with the structural model in Supp. Fig 2C. However, the E2-Ub isopeptide bond with E2-Lys141 does not appear well ordered in the view shown in Supp. Fig 2B. How did the authors confirm the linkage between Ub and E2?

When we performed E1-E2 thioester transfer assays with wild type Cdc34 under the same conditions used for the generation of the Cdc34^{A141K}-Ub(t) thioester mimetic, the product is fully DTT sensitive (please see Fig. 1b, left). This suggests that the native lysine residues in Cdc34 wild type do not have a tendency to form isopeptide linkages to Ub under our assay conditions and that A141K is the major linkage site in the thioester mimetic. Regarding electron density, as shown more clearly below the Ub(t) C-term tail is well ordered and in close proximity to Cdc34 Lys141. Although Cdc34 harbors several other lysine residues, they are distant from the E2 active site (shown below). To further address this issue, we assessed the stability of our Uba1/Cdc34-Ub(t)/Ub(a)/ATP•Mg²⁺ sample over a time frame of 15 days at 18 C which corresponds to the same conditions used for Uba1/Cdc34-Ub(t) crystal production and harvest. The results indicate that the Cdc34^{A141K}-Ub isopeptide bond does not appreciably degrade during course of the 15 day incubation period, even in the presence of DTT in the sample buffer. We now include this data in Supplementary Fig. 1c, right.

4) One important missing control for the E1 mutational analysis is to show that the E1 enzyme is not impaired for charging by Ubiquitin. Provide quantification of E1~Ub formation in Supp. Fig 3a.

We thank the reviewer for this suggestion and now present the E1~Ub thioester formation assays for the E1 mutants in Supplementary Fig. 4a. The results show that E1~Ub thioester formation activity for the panel of E1 mutants ranges from 86.7~103.2%, and thus the mutants are not significantly impaired for charging of Ub.

5) Do the 4 open state structures display similar crystal packing arrangements in the unit cell that could account for the positioning of Ub(t)? Do the 4 closed state structures display similar packing arrangements in the unit cell that could account for the positioning of Ub(t)?

If the packing arrangements for all 8 molecules in the asymmetric unit are very different, and yet only two conformations of Ub were observed, then that would provide further support to the notion that the conformations are more likely functionally relevant rather than coincidental crystal packing artifacts.

We thoroughly analyzed crystal packing patterns in our structure both manually and using the PISA server (<http://www.oecd.org/pisa/>). As shown in the figure below, although related, none of the four open state nor closed state structures display the same crystal packing arrangement. We also note that a similar Ub(t)^{OPEN}/FCCH domain interface was observed in the doubly loaded Uba1 structure determined in the absence of E2 (PDB: 4NNJ) in which Ub(t) is situated in a completely different crystal packing environment compared to the four Ub(t)^{OPEN} copies in our structure. Similarly, there are two structures of Cdc34 in which Ub interacts with a similar surface as Ub(t) closed (PDB: 4MDK and 6NYO) despite also having completely different crystal packing environments compared to the four copies of Ub(t)^{CLOSED} in our structure. These previous studies, together with the biochemical data we present in our manuscript strongly support the notion that the Ub(t) conformations observed in our structure are functionally relevant rather than coincidental crystal packing artifacts.

Uba1/Cdc34-Ub(t)^{OPEN}

Uba1/Cdc34-Ub(t)^{CLOSED}

Figure | Crystal packing environments around the eight copies of Uba1/Cdc34-Ub(t) in our structure.

Crystal packing environments around the four Uba1/Cdc34-Ub(t)^{OPEN} copies (*Top*) and the four Uba1/Cdc34-Ub(t)^{CLOSED} copies (*Bottom*) of the complex in our crystal structure. Uba1/Cdc34-Ub(t) copies are shown cartoons, while interacting molecules from symmetry mates are shown as ribbons. The structures are colored as in Fig. 2. The crystal packing environments are related but not identical for the different copies of the complex.

REVIEWERS' COMMENTS

Reviewer #1 (Remarks to the Author):

The authors have satisfied my concerns and improved their manuscript by addressing the issues raised during the review process.

Reviewer #3 (Remarks to the Author):

The authors have sufficiently addressed my comments.

As such, I now recommend publication of the manuscript in its current form.